# Bringing uncertainty quantification to the extreme-edge with memristor-based Bayesian neural networks

Djohan Bonnet[1,2] ✉, Tifenn Hirtzlin[1], Atreya Majumdar[2], Thomas Dalgaty [3], Eduardo Esmanhotto[1], Valentina Meli[1], Niccolo Castellani[1], Simon Martin[1], Jean-François Nodin[1], Guillaume Bourgeois[1], Jean-Michel Portal[4], Damien Querlioz [2] ✉ & Elisa Vianello [1] ✉

Safety-critical sensory applications, like medical diagnosis, demand accurate decisions from limited, noisy data. Bayesian neural networks excel at such tasks, offering predictive uncertainty assessment. However, because of their probabilistic nature, they are computationally intensive. An innovative solution utilizes memristors' inherent probabilistic nature to implement Bayesian neural networks. However, when using memristors, statistical effects follow the laws of device physics, whereas in Bayesian neural networks, those effects can take arbitrary shapes. This work overcome this difficulty by adopting a variational inference training augmented by a "technological loss", incorporating memristor physics. This technique enabled programming a Bayesian neural network on 75 crossbar arrays of 1,024 memristors, incorporating CMOS periphery for in-memory computing. The experimental neural network classified heartbeats with high accuracy, and estimated the certainty of its predictions. The results reveal orders-of-magnitude improvement in inference energy efficiency compared to a microcontroller or an embedded graphics processing unit performing the same task.

Hardware neural networks based on emerging non-volatile memories can bring intelligence to the edge at a very low energetic cost. In this context, filamentary memristors and phase change memories can be used in a very elegant and energy-efficient way. These devices can act as analog synaptic weights enabling neural network multiply-and-accumulate operations directly in memory by relying on Ohm's law and Kirchoff's current law[1–7]. Low-power systems of this kind could provide essential services: for example, medical devices could analyze patient measurements and detect life-threatening emergencies or automatically adjust treatment.

However, this vision comes with two major challenges. First, neural networks need to be programmed with precise weight values. Memristors and phase change memories are prone to conductance variability, instability, and drift[8–10], meaning that they act more as random variables than precise real weights, requiring the development of multiple program-and-verify and compensation techniques to correct their imperfection[11,12]. Second, conventional neural networks are not well suited to safety-critical applications, as they are notoriously bad at evaluating uncertainty[13–15]. When trained with small datasets, as is typically the case in medical applications, conventional neural networks tend to "overfit" training data and to provide highly certain answers in all situations[16,17].

More importantly, uncertainty in a machine learning context can have different origins, usually referred to as aleatoric (ambivalence between several known situations) and epistemic (unknown situation), which have different implications (see

[1]Université Grenoble Alpes, CEA, LETI, Grenoble, France. [2]Université Paris-Saclay, CNRS, Centre de Nanosciences et de Nanotechnologies, Palaiseau, France. [3]Université Grenoble Alpes, CEA, LIST, Grenoble, France. [4]Aix-Marseille Université, CNRS, Institut Matériaux Microélectronique Nanosciences de Provence, Marseille, France. ✉e-mail: djohan.bonnet@cea.fr; damien.querlioz@c2n.upsaclay.fr; elisa.vianello@cea.fr

"Results" section), and which conventional neural networks cannot tell apart[15,18].

Bayesian neural networks are an alternative class of neural networks, which have the potential to solve both challenges. In these networks, synaptic weights do not take on unique values but are instead represented by probability distributions[19–21], tracking the uncertainty about these weights. Bayesian neural networks are trained so that if we sample a value for each weight based on their probability distributions, we obtain a conventional neural network that constitutes a plausible interpretation of the training data. This contrasts with the training process of conventional networks, which solely fits the data, making them susceptible to overfitting[20]. For inference, we sample a pool of $M$ conventional neural networks from the Bayesian neural network and input the same data to all of them. Analyzing the output statistics enables not only a prediction but also a robust estimation of aleatoric and epistemic uncertainties[15].

The intrinsic randomness of memory nanodevices aligns naturally with the random variable nature of synapses in Bayesian neural networks. An implementation of Bayesian neural networks with memory nanodevices can be achieved by programming a neural network $M$ times to reproduce the sampling operation necessary to derive $M$ conventional neural networks from the Bayesian one. However, a critical question remains: how can we train Bayesian neural networks to align with the characteristics of memory nanodevices?

Synaptic weight probability distributions in Bayesian neural networks can take any shape[19,22], but the statistical properties of memristors and phase change memories follow rigid physics rules[23–25]. Filamentary memristors, for instance, demonstrate broader probability distributions at higher resistance states and narrower ones at lower resistance states[8,26], correlating resistance mean value and standard deviation. To overcome this difficulty, two recent studies proposed new devices with tunable inherent resistance probability distributions, using two-dimensional materials[27] and magnetic devices[28]. These solutions, were validated with simulations of Bayesian neural networks.

In the main contribution of our paper, we propose a dedicated technique for Bayesian neural networks—variational inference augmented by a "technological loss" that leads to networks readily implementable with more conventional memory nanodevices. Standard variational inference trains the mean value and standard deviation of each synapse via backpropagation to identify plausible interpretations of the training data. During the training process, the mean value and standard deviation of a synaptic weight evolve following different gradient values and become fully decorrelated. Our added technological loss constrains these synaptic weights and standard deviations to domains implementable with the selected nanodevices for a given technological implementation. We demonstrated this technique's effectiveness using standard nanodevices (filamentary memristors) to accomplish the first complete nanodevice-based Bayesian neural network implementation for a real-world task—classifying types of arrhythmia recordings with precise aleatoric and epistemic uncertainty. Our system utilized 75 arrays of fabricated $32 \times 32$ memristor chips integrating hafnium oxide memristors and CMOS peripheral circuitry for in-memory computations based on Kirchoff's laws.

In addition to the "technological loss", used here for the first time and critical for the success of our experiment, our work uses two additional techniques to overcome the problems related to the correlation of statistical properties of memristors already proposed in the literature. The specific choice of variational inference[20], also present in[27–29], ensures statistical independence between synapses. A prior simulation study[26] suggested Markov Chain Monte Carlo training, leading to statistically-dependent synapses, which is extremely challenging to realize experimentally due to memristor imperfection (see Supplementary Note 10). Second, each synaptic weight value is implemented using two memristors that are programmed independently, allowing the partial decorrelation of mean values and standard deviations of synaptic weights (see Result section and Supplementary Note 7). This idea was previously presented in several simulation studies[27,28,30]. Supplementary Note 14 presents an in-depth comparison of our work with the state of the art.

Moreover, our technological loss approach is generic. Using a different expression for it, we show that the same technique can be applied to phase change memory (by following a hybrid experimental-simulation methodology, employing a fabricated array of 16,384 phase change memories).

Our research is situated within a current trend of linking Bayesian concepts and nanodevice characteristics. A number of studies have identified connections between Bayesian models other than Bayesian neural networks and nanodevices. The Bayesian machine of Ref. [31] uses memristors as memory for the model parameters and uses stochastic computing to perform inference on a Bayesian network. Bayesian networks differ from Bayesian neural networks. The former are constructed using expert knowledge and are fully explainable, which makes them ideal for tasks like sensor fusion. On the contrary, Bayesian neural networks are trained from the ground up and excel on more data-intensive tasks like electrocardiogram or electroencephalogram classification. From a circuit point of view, the Bayesian machine also differs strongly from the present work: The Bayesian machine is a digital system that tolerates memristor imperfections but does not exploit them[31]. Several studies also suggested leveraging the stochastic traits of nanodevices to facilitate Bayesian network inference[32–35]. Moreover, Ref. [23] exploited the probabilistic nature of memristors to perform Bayesian learning. This approach can only be applied to small-scale tasks. Unlike Bayesian neural networks, it does not suffer from the limitations imposed by the correlation of the mean value and standard deviation of memristors, which makes it more straightforward to implement[23]. Finally, the work of Ref. [36] treats memristors as Bayesian variables and uses them to program deterministic, non-Bayesian neural networks in order to increase hardware resilience.

In this paper, we introduce and describe the general architecture of our Bayesian neural networks, which are based on memory devices, and our technique to match imperfections in nanodevices with the probability distributions of a Bayesian neural network. We then show the experimental classification of arrhythmia with a proper uncertainty evaluation, using 75 arrays of filamentary memristors.

## Results
### Memory devices-based Bayesian neural networks
For our experiments, we considered a two-layer Bayesian neural network (Fig. 1a, b), trained to differentiate nine classes of heart arrhythmia from electrocardiogram (ECG) recordings[37]. While in Artificial Neural Networks (ANNs), the synaptic weights are point estimates, Bayesian neural networks replace them with probability distributions. A natural way to implement a Bayesian neural network with memristors or phase change memories is to use a collection of $M$ distinct memory arrays to represent each layer of the neural network (Fig. 1c, d). We sample $M$ weight values for each synapse based on its probability distribution in the Bayesian neural network, and we program them to the $M$ memory array. We will see in the next sections that the inherent probabilistic effects in memory devices allow us to perform the sampling and programming operations simultaneously, transforming the largest drawback of emerging memory devices, their variability, into a feature. This approach leads to a collection of $M$ independent in-memory neural networks. By presenting the same input to each of these arrays, we obtain a collection of $M$ different outputs representing the output distribution of these neural networks.

The benefit of using distributions, instead of deterministic values, is that we can quantify the uncertainty of the neural network's output.

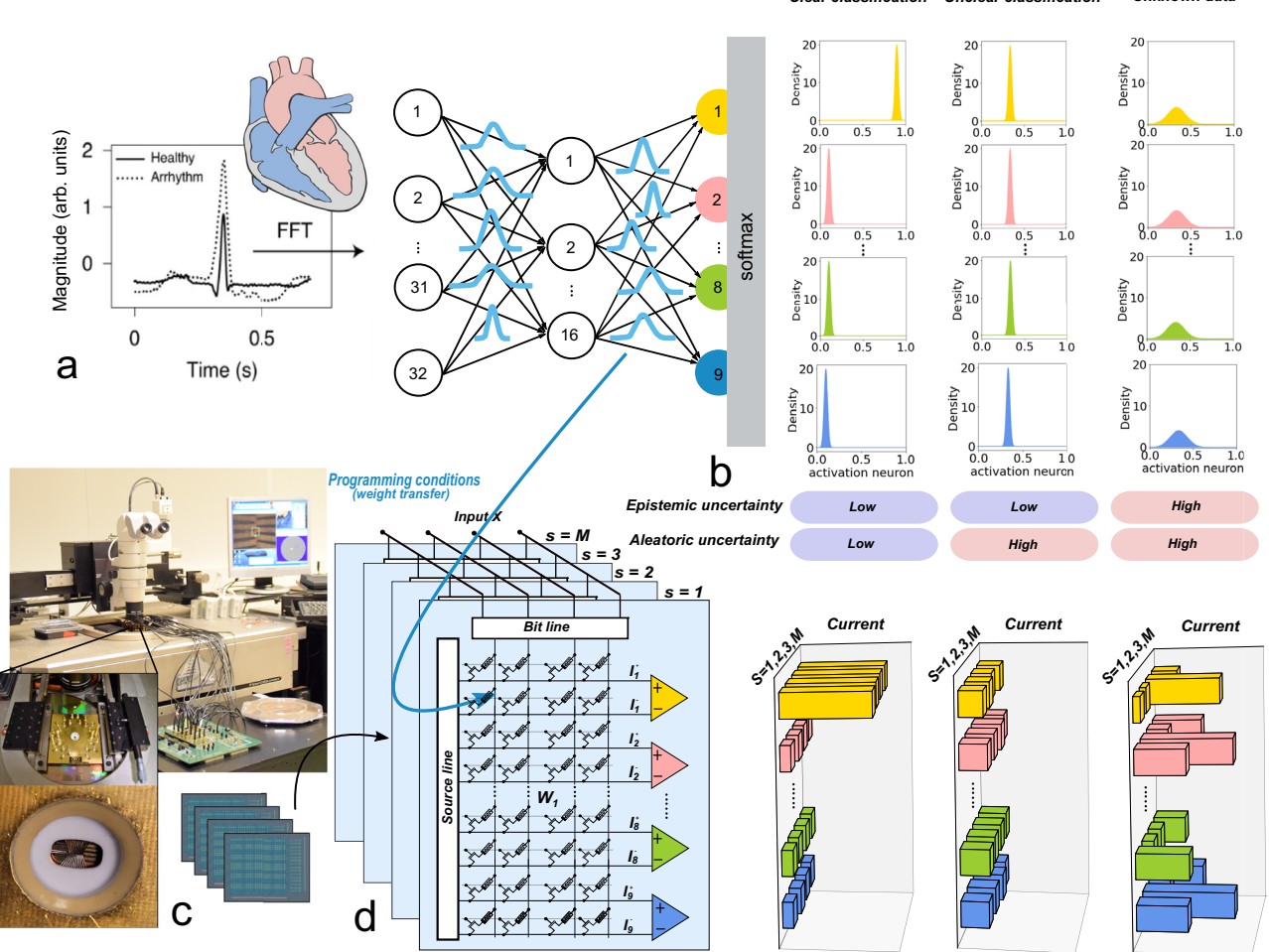

**Fig. 1 | General architecture of the Bayesian neural network. a** Schematic of the Bayesian neural network used for heart disease (arrhythmia) classification. In Bayesian neural networks, the weights are represented by probability distributions, thus naturally including uncertainty in the model. The heart schematic has been reproduced from the work of Patrick J. Lynch, medical illustrator; C. Carl Jaffe, MD, cardiologist. https://creativecommons.org/licenses/by/2.5/. **b** Example of output neuron activation distributions, obtained for certain output, uncertain output, due to noisy input data, and unknown data (i.e. out-of-distribution data). **c** Experimental setup. **d** Hardware implementation of a Bayesian neural network by combining multiple versions of ANNs.

Intuitively, each of the $M$ independent neural networks constitutes a reasonable hypothesis interpreting the data used to train the Bayesian neural network. The spread of the output distributions captures the certainty or lack of certainty in the model's predictions. A unique property of Bayesian neural networks is their ability to differentiate between aleatoric and epistemic uncertainty. To understand the appeal of this feature, we can use the example of a medical device trained to recognize different types of arrhythmia (irregular heartbeats) in a patient. Aleatoric uncertainty characterizes situations where a measurement could be consistent with different types of arrhythmia. Epistemic uncertainty, on the other hand, arises when an arrhythmia type that is significantly different from the data used to train the device appears. Recognizing and differentiating those two types of uncertainty is essential, as they could be indicating that the patient's condition has evolved[18,38].

Uncertainty can be quantified by analyzing both the mean and the variance values of the output neuron activations, as illustrated in Fig. 1b–d. If an output is highly certain (clear classification), all $M$ neural networks will have the same active output neuron (value close to one), and inactive output neurons (value close to zero), and both the aleatoric and epistemic uncertainties will be low (see Methods for the mathematical definitions of aleatoric and epistemic uncertainties). If measurement imprecision causes ambivalence in the predictions

(unclear classification), e.g., the features allowing differentiation of the arrhythmia types are lost in noise, none of the $M$ neural networks will provide a certain prediction: all of them will hesitate between the same classes, with non-zero outputs on these classes. The variance of the output remains low, as all $M$ networks have a consistent behavior: aleatoric uncertainty increases, whereas epistemic uncertainty does not. But what happens if a new arrhythmia type that was not present during training appears, and the network has to classify it? This is an example of out-of-distribution test data (case Unknown data in Fig. 1b–d). All $M$ neural networks sampled from the Bayesian neural network will tend to make a different interpretation of this unknown data resulting in a high variance in the distribution of output neurons. Both aleatoric and epistemic uncertainty will be high. Classifying out-of-distribution data, therefore, is possible by measuring output distributions. This is of fundamental importance for safety-critical applications like medical diagnoses or autonomous driving.

Performing inference with our approach requires massive parallel Multiply-and-Accumulate (MAC) operations. These operations are power-hungry when carried out on CMOS-based ASICs and field-programmable gate arrays, due to the shuttling of data between processor and memory. In this work, we use crossbars of memristors that naturally implement the multiplication between the input voltage and the probabilistic synaptic weight through Ohm's law, and the

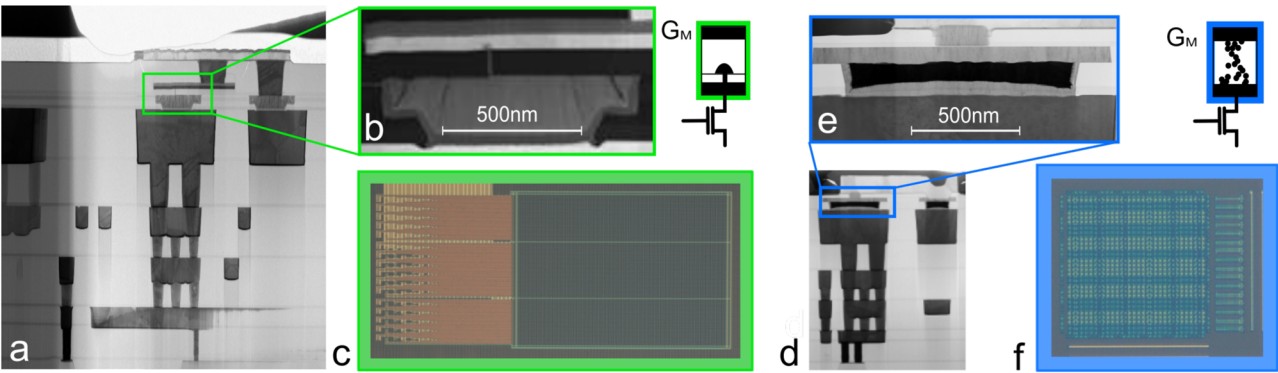

**Fig. 2 | Fabricated filamentary memristor and phase-change memory based array die. a** Transmission electron microscopy image of a phase-change memory in the back end of line of our hybrid memristor/CMOS process. **b** Transmission electron microscopy image of a phase-change memory. **c** Optical microscopy photograph of the phase-change memory-based 1T1R array. **d** Transmission electron microscopy image of a filamentary memristor in the back end of line of our hybrid memristor/CMOS process. **e** Transmission electron microscopy image of a filamentary memristor. **f** Optical microscopy photograph of the filamentary memristor-based 1T1R array.

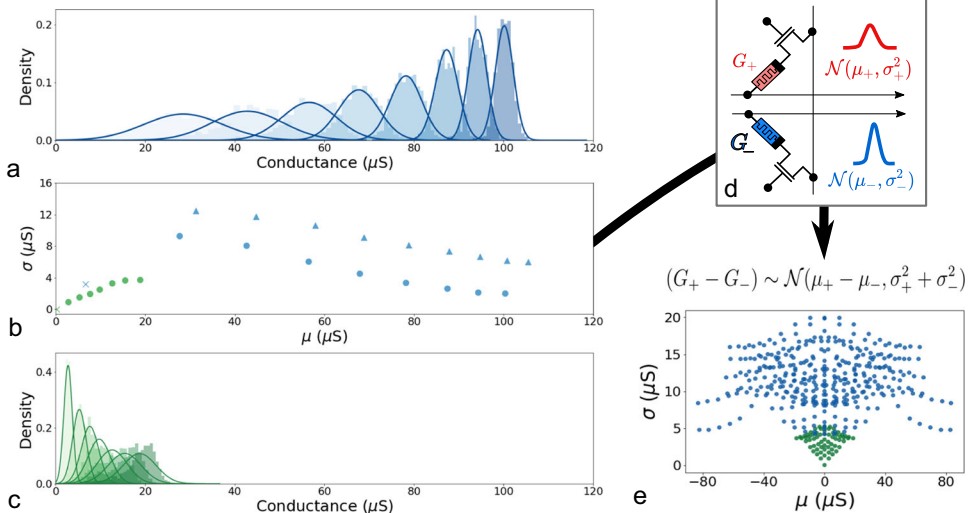

**Fig. 3 | Filamentary memristor and phase-change memories as physical random variables with normal distribution. a** Probability densities of 2,048 filamentary memristors programmed with eight different programming current values. **b** Domain of the Gaussian distributions experimentally achieved exploiting different programming conditions for filamentary memristors (blue) and phase-change memories (green). Triangles represent one-shot programming, dots represent iterative programming and the cross represents the low conductance state. **c** Probability densities of 2,048 phase-change memories programmed with seven different programming current values. **d** Schematic of the proposed synaptic circuit. Each sample of a Bayesian probabilistic weight is stored as the difference between the conductance values of two adjacent memory cells. **e** Domain of the normal distributions (Γ) that can be experimentally obtained exploiting the circuit in **d** by storing samples on two memory cells.

accumulation through Kirchhoff's current law[2,7,39,40], to significantly lower power consumption.

## Filamentary memristor and phase-change memory as normal distributions

Among the non-volatile memories that can be integrated in advanced commercial processes, phase-change memories (PCM) and filamentary memristors have been widely studied for analog in-memory neural network implementation, because of the possibility of adjusting the conductance level of these devices. In our previous work, we demonstrated that the intrinsic variability in filamentary memristors can be leveraged to store the probabilistic weights of Bayesian neural networks[26]. However, the conductance distribution follows strict rules due to device physics: the mean value, $\mu$, and the standard deviation, $\sigma$, are strongly correlated. Phase-change memories suffer from the same limitation[24]. Bayesian neural networks require a larger space of normal distribution with mean values that are uncorrelated with the standard deviations. Here, we come up with a new synaptic circuit and the associated programming strategy to obtain largely unrestricted $\mu$ and $\sigma$ values. To illustrate the practicality of the proposed solution, we fabricated and tested arrays of hafnium-oxide-based filamentary memristors and of germanium-antimony-tellurium phase-change memories in a one-transistor-one-resistor (1T1R) configuration (Fig. 2). Both memory technologies have been integrated into the back end of line (BEOL) of a 130-nanometer foundry CMOS process with four metal layers (see Methods). Figure 3a, c shows the distributions of 2048 filamentary-based memristors and phase-change memories, respectively, programmed in eight conductance levels.

In both cases, the standard deviation of the distribution is related to its mean value and cannot be chosen independently. The resulting domain of normal distributions that can be achieved by exploiting device variability ($\sigma$) is thus bounded to a one-dimensional space for both technologies (Fig. 3b). In filamentary memristors, $\sigma$ decreases for increasing conductance values due to the Poisson-like spread of the

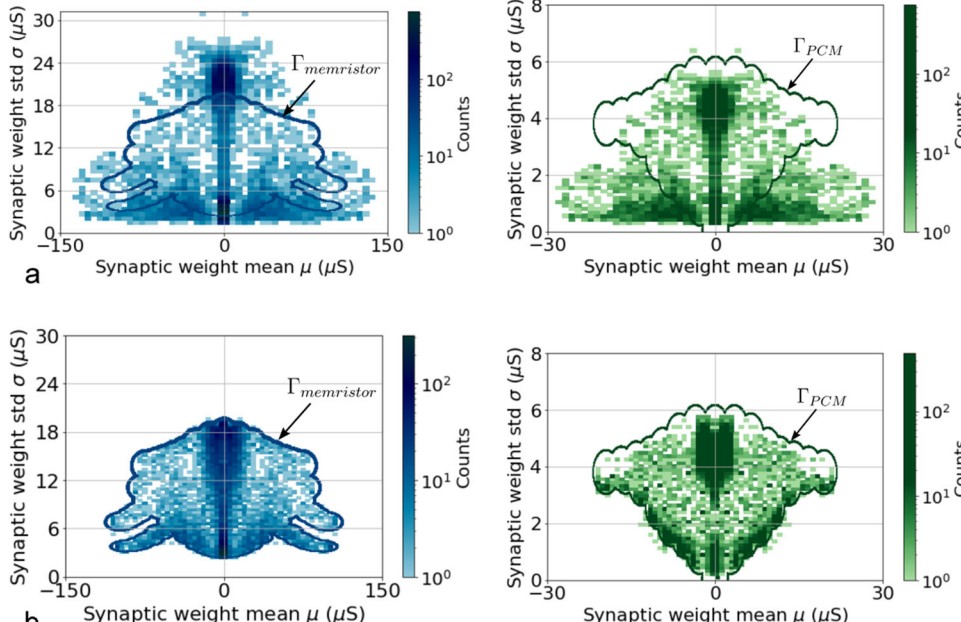

**Fig. 4 | Domains of normal distribution obtained with classical Variational Inference and the proposed technologically plausible method. a** Domain of normal distributions $\theta=(\mu, \sigma)$ for the synaptic weights obtained after training our reference arrhythmia classification task with the classical variational inference method and mapping the software values to the conductance range achievable with filamentary memristors, $\Gamma_{memristor}$ (blue) and phase-change memories, $\Gamma_{PCM}$ (green). Weight values have been scaled to a conductance value (expressed in microsiemens) using a factor calculated to minimize the statistical distance between the normal distributions calculated by software and the available experimental ones (see Methods). **b** Domain of normal distributions obtained after training employing the new technologically calibrated method on filamentary memristors (blue) and phase-change memory experimental data (green).

number of defects injected during the programming operation[41]. In phase-change memories, the trend is inverted: device variability increases with the conductance values, moving from a full to a partial amorphous material[24,25]. For both technologies, the standard deviation can be reduced by adopting an iterative program-and-verify scheme (see Methods). To extend the domain of normal distributions, we store each sample of a probabilistic weight as the difference between the conductance values of two adjacent memory cells, as shown in (Fig. 3d). This method is particularly useful, because the difference between two normal distributions is still a normal distribution. Figure 3e illustrates the corresponding technologically-plausible domain of normal distributions for both filamentary memristors ($\Gamma_{memristor}$) and phase-change memories ($\Gamma_{PCM}$). Both filamentary memristors and phase change memories suffer from conductance instability over time, due to the local recombination of oxygen vacancies and structural relaxation of the material, respectively[11,42]. However, the shape of the technologically plausible domain of normal distributions is only slightly altered by these effects (see Supplementary Notes 4, 5, 6).

### Hardware-calibrated training
In Bayesian neural networks, the weights are probability distributions, given by the posterior probability distributions, $p(\mathbf{\Omega}|\mathbf{D})$, where $\mathbf{D}$ is the training data. The most popular methods to approximate the posterior distributions are Markov Chain Monte Carlo (MCMC) sampling[43] and variational inference (VI)[44]. We proposed the transfer of a Bayesian neural network trained by MCMC, an algorithm that samples the posterior exactly, in our previous work[26]. However, MCMC lacks scalability and its training time is orders of magnitude longer than that of variational inference[14]. MCMC methods typically require a huge number of samples to approximate the posterior, involving high memory density to store it, rendering them area and energy inefficient. Moreover, the mapping of the software posterior on hardware causes a loss in accuracy and estimation of both epistemic and aleatoric uncertainties of several percentage points (see Supplementary Note 10).

Here, we use the variational inference method, which scales better than MCMC[19]. Rather than sampling from the exact posterior, the latter is approximated with normal distributions, $q(\mathbf{\Omega}|\boldsymbol{\theta})$, where $\boldsymbol{\theta}$ represents the mean and standard deviation ($\boldsymbol{\mu},\boldsymbol{\sigma}$). The estimation is performed by minimizing the loss function, the Kullback-Leibler divergence between $p(\mathbf{\Omega}|\mathbf{D})$ and $q(\mathbf{\Omega}|\boldsymbol{\theta})$:

$$Loss_{VI} = KL[q(\mathbf{\Omega}|\boldsymbol{\theta})||p(\mathbf{\Omega}|\mathbf{D})]. \qquad (1)$$

During the training phase, for each weight, $\mu$ and $\sigma$ are learned using the backpropagation algorithm (see Methods). Figure 4a illustrates the domain of the normal distributions $\theta = (\mu, \sigma)$ of the synaptic weights obtained after software training our reference arrhythmia classification task and mapping the software values to the conductance range achievable with filamentary memristors (blue) and phase-change memories (green). The mapping operation is a linear scaling of $\theta = (\mu, \sigma)$ by a factor $\gamma$ calculated to minimize the statistical distance between the normal distributions calculated by software and the available experimental ones (see Methods). However, this operation is not sufficient to meet the technology requirements: the desired domain exceeds the available experimental one for both filamentary memristors ($\Gamma_{memristor}$) and phase-change memories ($\Gamma_{PCM}$). To compel the learned normal distributions to match with the hardware experimental electrical characteristics, we imposed that $\theta$ belong to the experimental $\Gamma$ domain by adding the "technological loss" term to the loss function during the training process:

$$Loss = Loss_{VI} - \log(U_\Gamma(\boldsymbol{\theta})), \qquad (2)$$

where $U_\Gamma(\boldsymbol{\theta})$ is designed to approach uniform distribution over the $\Gamma = \Gamma_{memristor}/\gamma$ or the $\Gamma = \Gamma_{PCM}/\gamma$ domain (see Methods). Figure 4b illustrates the effectiveness of the proposed hardware-calibrated training method: the normal distributions obtained by software simulations perfectly map on both phase-change memories and

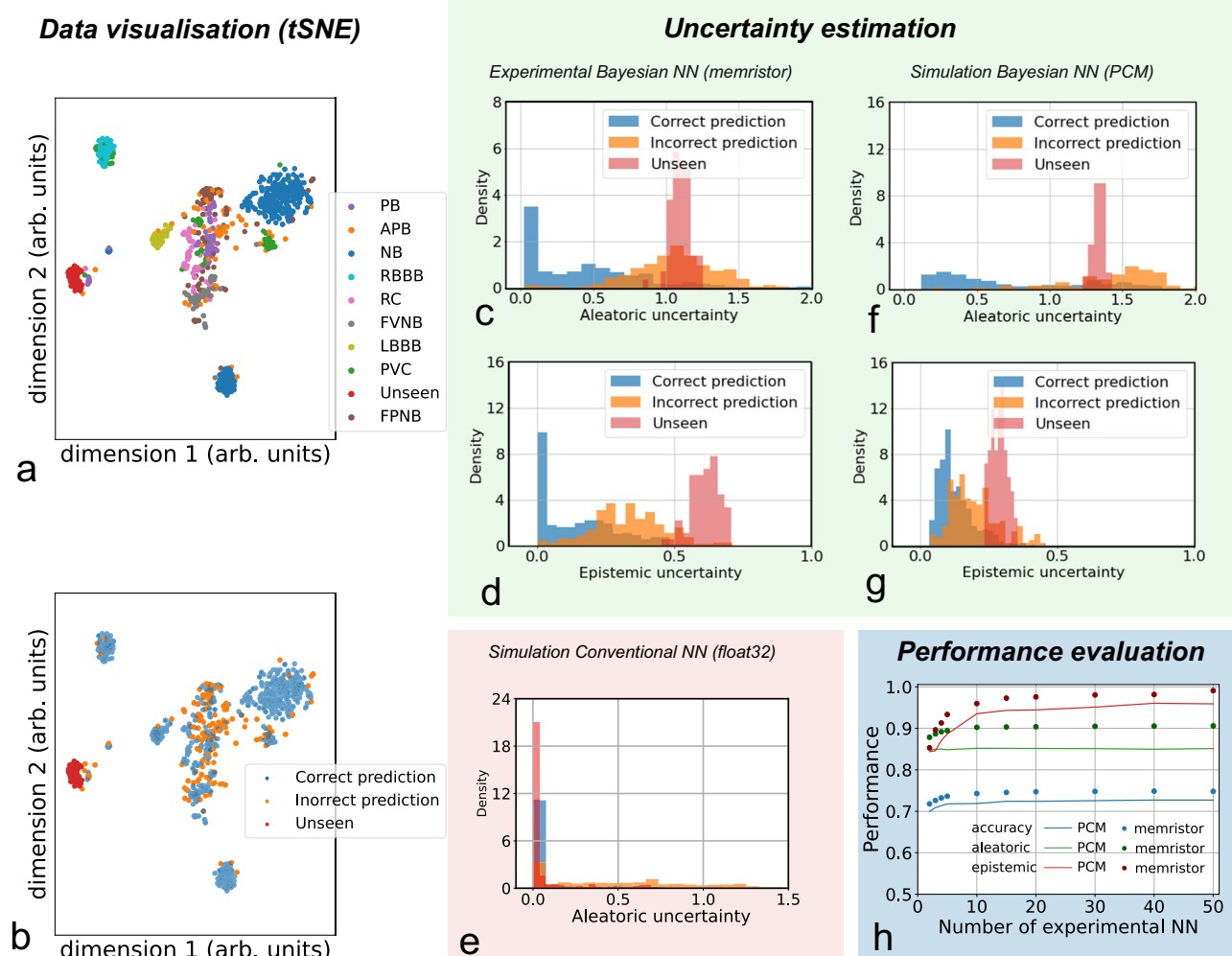

**Fig. 5 | Measurements of the fabricated memristor-based Bayesian neural network and simulations of a PCM-based Bayesian neural network. a** tSNE visualization of input data, different colors representing different classes (diseases). Nearby points correspond to similar data and distant points to dissimilar data. **b** tSNE visualization of experimental data classification. The different colors represent points correctly or incorrectly predicted and unseen data. **c** Experimental probability density distribution of the aleatoric uncertainty for correct predictions, incorrect predictions and unseen diseases, using filamentary memristors. **d** Experimental probability density distribution of the epistemic uncertainty for correct predictions, incorrect predictions and unseen diseases, using filamentary memristors. **e** Simulated probability density distribution of the aleatoric

filamentary memristors experimental values. We, therefore, demonstrated that by taking hardware physics into account while developing the training algorithm, it is possible to make variational inference a technologically plausible algorithm.

### Experimental uncertainty estimation

To validate our approach, we programmed a Bayesian neural network, trained to recognize arrhythmia, onto a collection of the filamentary memristor dies (Fig. 2f). We classified ECG diagnosis beats using a two-layer Bayesian neural network featuring 32 inputs, 16 hidden neurons in the first layer, and nine output neurons in the second layer. We trained the Bayesian neural network, using the filamentary memristor technological loss, on nine classes: healthy beat and eight types of arrhythmias. During testing, we added a tenth class corresponding to a non-previously seen type of arrhythmia. Following the architecture presented in Fig. 1, we programmed $M = 50$ independent realizations of model parameter vector $\boldsymbol{\theta}$, representing the learned posterior $q(\boldsymbol{\Omega}|\boldsymbol{\theta})$,

uncertainty for correct predictions, incorrect predictions and unseen diseases for a conventional neural network with the same architecture and using *float*32 encoding for the synapses. **f** Simulated probability density distribution of the aleatoric uncertainty for correct predictions, incorrect predictions and unseen diseases, using PCMs. **g** Simulated probability density distribution of the epistemic uncertainty for correct predictions, incorrect predictions and unseen diseases, using PCMs. **h** Measured (memristor) and simulated (PCM) accuracy, epistemic uncertainty, and aleatoric uncertainty performance (calculated as the area of the ROC curves presented in Suppl. Note 1) as a function of the number of pair of devices per synapse.

i.e., we transferred each model realization into an array of conductance values. Each realization can fit in 1.5 dies presented in Fig. 2f (one die for the first layer of the neural network, and a half die for the second layer, see Methods), and described in detail in Supplementary Note 13; therefore, we needed to program a total of 75 dies. Multiply-and-accumulate operations were performed directly in memory using Ohm's and Kirchoff's law (see "Methods" and Supplementary Note 13). Activation functions were calculated in software. The array performed all the multiply and accumulate operations needed to classify 1000 beats in the test data set. Supplementary Note 16 recapitulates the different steps of our experiment, from training to inference.

Figure 5 presents the electrical characterization results of the memristor-based Bayesian neural network. To visualize the input data, we used the t-distributed stochastic neighbor embedding (t-SNE) statistical method (Fig. 5a, b). This visualization technique represents each high-dimensional input data by a point in a two-dimensional space, in a way that similar data correspond to nearby points and

**Table 1 | Comparison of accuracy and uncertainty prediction performances**

|  | Conventional ANN (*float*32) | Bayesian (*float*32) | Bayesian Hardware (filamentary memristor experimental) | Bayesian Hardware (filamentary memristor simulation) | Bayesian Hardware (phase-change memory simulation) |
|---|---|---|---|---|---|
| Accuracy classification | best: 81% mean: 80% | best: 80% mean: 79% | 75% | best: 76% mean: 76% | best:73% mean: 73% |
| Prediction confidence (aleatoric) [AUC] | best: 0.90 mean: 0.79 | best: 0.92 mean: 0.90 | 0.91 | best:0.91 mean: 0.89 | best:0.85 mean: 0.87 |
| Anomaly detection (epistemic) [AUC] | 0.5 | best: 1 mean: 0.95 | 0.99 | best: 0.96 mean: 0.92 | best: 0.96 mean: 0.82 |

distant points represent dissimilar data. Figure 5a illustrates the two-dimensional projections of the input data used during inference on the test dataset. The data belonging to a given class (disease) display a cluster. The unseen diseases (i.e., beats that do not belong to a class learned in the training phase) are the red points. Figure 5b uses the same representation, where the colors represent data points correctly (blue) or incorrectly (orange) classified by our experiment, while the unseen disease data points are plotted in red. Our experiment recognizes 75% of the data points correctly. Most errors concern points that lie at the border between several clusters in the t-SNE plot, suggesting that they might be ambivalent (high aleatoric uncertainty cases). To investigate this idea further, Fig. 5c shows the measured probability density distributions of the measured aleatoric uncertainty, which provides a measure of the confidence of network prediction. The different colors represent correct predictions (blue), incorrect predictions (orange), and unseen data (red). The aleatoric uncertainty is lower than 0.5 for 62% of all correctly classified data points, while it is higher than 0.5 for 97% of all incorrectly classified data points and unseen disease data points. This result means that our experiment correctly determined as uncertain all of its errors and the unseen disease. It also flagged as uncertain some of its correct predictions, which is expected, as some of them might be ambivalent cases.

The situation is quite different when we look at the measured epistemic uncertainty (Fig. 5d). In total, 97% of all correctly and incorrectly classified data points have an epistemic uncertainty lower than 0.5. Conversely, 98% of the unseen disease data points have epistemic uncertainty higher than 0.5. These results mean that experiments can differentiate ambivalence between classes from the presentation of new unknown inputs.

These results come in sharp contrast with those of a simulated conventional neural network with the same architecture. This type of neural network, by construction, has no epistemic uncertainty, and the aleatoric uncertainty tends to be extremely low whatever the input (Fig. 5g). This overconfidence is due to the small size of our dataset, making conventional neural networks particularly prone to overfitting.

The dies that we used for phase-change memory characterization (Fig. 2c) are conventional memory arrays that do not allow in-memory computing and cannot be implemented as a full in-memory Bayesian neural network, unlike what we achieved for filamentary memristors. Therefore, we used our extensive statistical measurements of phase change memories (Fig. 4) to simulate such a network, using the simulator validated in Supplementary Note 2. Figure 5e, f, plotted using the same methodology as Fig. 5c, d, shows the same features as observed in the experimental memristor-based system.

The selection of the threshold value for uncertainty quantification depends on the specific application and context. To push the interpretation of our experimental results further and make an in-depth assessment of the capability of our experiment to evaluate uncertainty, we used receiver operating characteristic (ROC) curves, a widely used metric for diagnostic ability, obtained by plotting the true positive rate as a function of the false positive rate for various threshold settings. A perfect classifier would yield the (0, 1) point, i.e., an area under the curve (AUC) of one, corresponding to no false negatives and no false positives. The ROC curve of a random classifier approaches the diagonal line, i.e., an area under the curve of 0.5. We first compute the ROC curve corresponding to the differentiation between correct predictions and incorrect predictions, based on aleatoric uncertainty. Its area under curve provides a measure of the performance of a network in terms of aleatoric uncertainty evaluation. We also compute the ROC curve corresponding to the differentiation between known and unknown data, based on epistemic uncertainty. Its area under curve provides a measure of the performance of a network in terms of epistemic uncertainty evaluation. Table 1 lists the raw accuracy and the aleatoric and epistemic evaluation performance for our experiment, a purely software version of the Bayesian neural network programmed in our experiment, and a conventional neural network with the same architecture. For all simulated results, the training process was repeated ten times and we reported both the mean and best performance. Corresponding ROC curves, as well as methodological details, are presented in Supplementary Note 1. We should remark that an area under the curve of one is not expected in the case of aleatoric uncertainty: predictions with low aleatoric certainty are sometimes correct.

Our experiment nearly matches the best performances obtained by a software neural network in terms of both aleatoric (area under the curve of 0.91) and epistemic uncertainty evaluation (area under the curve of 0.99), with a small reduction in terms of raw accuracy. This result shows the impressive robustness of our approach, which fully embraces the imperfections of experimental results. The conventional, non-Bayesian neural network exhibited no capacity to recognize unknown data and has the minimum epistemic uncertainty evaluation performance (0.5). In terms of aleatoric evaluation, it has, on average, reduced performance with regard to Bayesian networks, highlighting again the overconfidence of such networks. Still, the best one approaches the performance of Bayesian networks on our arrhythmia detection task. Table. 1 also includes results for a simulated Bayesian neural network based on phase change memories. They suggest that this network would function equivalently to the filamentary memristor-one, with a slight reduction in terms of accuracy and uncertainty evaluation (expressed by the area under the two ROC curves).

A drawback of our approach with regard to conventional neural networks is that we need several (M) versions of the neural networks. This number, however, does not necessarily need to be high. Figure 5h shows its effect on the Bayesian neural network accuracy and on its capability to evaluate uncertainty, measured by the area under the curve of the two ROC curves mentioned above, obtained using aleatoric and epistemic uncertainty. The accuracy and aleatoric area under the curve approach their saturation values with ten neural networks. The epistemic area under the curve takes a higher number of implementations to converge; however, with ten neural networks, it reaches 0.96, close to its maximum value (0.99).

## Discussion

This work demonstrates experimentally a simple and energy-efficient realization of a Bayesian neural network by directly storing the probabilistic weights into resistive memory-based crossbar arrays. The device variability in both filamentary-based memristors and phase-

change memories is used to store physical random variables that sample analog conductance values from normal distributions with re-configurable mean and standard deviation. The Bayesian neural networks are trained following a special variational inference approach, incorporating a "technological loss" to overcome the hardware limitations linked to the device physics. We implemented a whole network using a collection of filamentary memristor arrays allowing in-memory computing. The resulting Bayesian neural network matches software simulations in terms of accuracy, and in terms of aleatoric and epistemic uncertainty evaluations, as evidenced by ROC curves for the identification of misclassified heartbeats and unknown data heartbeats.

The critical element for this approach to succeed was the inclusion of the technological loss in the training process to correct for the constraints of technology. Supplementary Note 9 shows that without using the use of this loss, the memristor-based Bayesian neural networks would have performed poorly. The fact that Bayesian neural networks based on phase change memory still achieve almost matching performance and uncertainty evaluations with the memristor case also demonstrates the power of the technological loss term. Figure 4 shows that the mean value/standard deviation space that can be programmed on phase change memories is more skewed than that of filamentary memristors: it is impossible to program synapses with low standard deviation and high mean weight magnitude on phase change memories. The technological loss ensures that the resulting Bayesian neural network still performs well.

Another important aspect is the use of two devices per weight value to decorrelate mean value and standard variation of weights. Supplementary Note 7 shows that our approach would not have been successful using a single device per weight sample. Using more than two devices can decorrelate the statistical properties of the synapses further; however, Supplementary Note 7 shows that using four devices instead of two brings limited benefits in terms of Bayesian neural network performance, despite the high hardware cost.

In our work, we employed two programming techniques (with and without iterative properties) to extend the domain of possible statistical properties of the synapses when using filamentary memristors. Supplementary Note 8 confirms that this choice improved the performance of the programmed Bayesian neural network, but that acceptable performance could have been obtained using a single programming technique. For phase change memory, iterative programming is fundamental to programming the devices and is used in all cases.

The most important limitation of our approach is that it requires the use of multiple pairs of devices per synapse to represent a distribution of its synaptic weight. The results of Fig. 5h show that the number of samples per synapse does not need to be large. Bayesian neural networks excel in relatively small-data regimes, where strong uncertainty is present: they are not large networks, making device overhead bearable. Currently-developed resistive memories integrated in three dimensions may be particularly suitable to our architecture, which features multiple devices per synapse[45].

Long-term drift and instability effects in the conductance of memory devices is a major limitation of analog neural networks based on nanodevices, particularly phase change memories. Supplementary Notes 5 and 6 reveal an impressive resilience of our approach to these effects, which can be tied to the fact that it does not require a precise value for device conductance, but rather values representative of a probability distribution.

This study is based on a relatively simple arrhythmia classification dataset, small enough to be implementable on our in-memory computing memristor arrays. We intentionally chose a dataset with high ambivalence, leading to a relatively low accuracy and uncertainty. On the similar but clearer dataset of ref. 46, the simulator validated in Supplementary Note 2 suggested that our approach would reach an

accuracy of 94%. Finally, Supplementary Notes 11 and 12 show that our approach can be applied to evaluate uncertainty in situations with larger machine-learning data sets (MNIST handwritten digit recognition and CIFAR image recognition).

Two principal methods exist for estimating uncertainty in non-Bayesian artificial neural networks (ANNs). The first, deep ensembles, trains multiple identical ANNs, creating a prediction distribution but offering no energy or hardware benefits[47]. Moreover, implementation challenges arise when transferring high-precision parameters into the imprecise conductance states of resistive memory in memristor-equipped ANNs. In contrast, Bayesian neural networks adeptly exploit memristor variability to store random variables, making them ideal for resistive memory-based hardware. The second method, Monte Carlo dropout, generates a prediction distribution by randomly disabling nodes within the model[48,49]. However, the requirement for multiple forward passes precludes any reduction in energy consumption, and it is not natural to implement in a memristor-based circuit. Besides these general techniques, some task-specific approaches have also been proposed. In particular, some models use artificial neural networks representing Gaussian distributions, where one neuron represents the mean and another the standard deviation of a distribution. This approach has proven useful in tasks such as detecting out-of-distribution data in video surveillance or improving the accuracy of bounding box regression[50–53]. Bayesian neural networks constitue a more general solution to the uncertainty evaluation challenge.

Our in-memory computing chip embeds only parts of the peripheral circuits to implement a full neural network (see Supplementary Note 13), with the remaining functions implemented on printed circuit board. In particular, analog-to-digital-converters, typically the largest source of energy consumption in analog compute-in-memory systems[3,7,39] are implemented off-chip in our experiments. For this reason, to estimate the inference energy consumption of a final in-memory Bayesian neural network, we relied on numbers obtained in industrial in-memory computing platforms platform[3,7,39]. We found a cost ranging between 0.7 and 2.5 nanojoules per inference (see Supplementary Note 15). We do not expect a significant difference between filamentary memristors and phase change memory-based systems, as analog-to-digital-converters and word line charging, the two most power-consuming operations analog compute-in-memory systems, have an energy consumption that is independent of the resistance of the memory elements. As a comparison, following a methodology introduced in ref. 31 we implemented the computation of our reference Bayesian neural networks on a microcontroller unit typically used for edge AI applications. (The Methods sections provide the details and the limitations of this comparison between an emerging and an established technology.) This control experiment consumed 170 microjoules per inference. We also implemented the same computations on a low-power embedded graphics processing unit (GPU) board (NVIDIA Jetson Nano), widely used for edge AI applications requiring more computational power. GPUs excel at parallel computing: they can compute the inference on all samples corresponding to the same input simultaneously. They can also process multiple inputs simultaneously (batching). We found experimentally (see Methods) that the energy consumption depends quite dramatically on the degree of batching, which, depending on the embedded application, may or not be exploitable. When processing a single input, the GPU uses an energy of 80 microjoules (1300 microjoules for the entire board). When processing 100 inputs simultaneously, the GPU uses an energy of 9 microjoules per input (35 microjoules for the entire board). The efficiency of our approach suggests that Bayesian neural networks can be used at the edge in extremely energy-constrained systems, such as medical devices, where reliable decisions are needed.

## Methods

### Filamentary and phase change memory technology and circuits

The circuits described in the Results section were fabricated using a low-power foundry 130-nanometers process with four metal layers. Both phase-change memories and filamentary memristors were fabricated on tungsten vias in metal layer four. The filamentary memristors consist of a 5-nanometer thick metallic bottom electrode, a 5-nanometer thick $HfO_x$ active layer deposited by atomic layer deposition, and a 10-nanometer thick Ti top electrode. The memory element is fabricated as a mesa structure with a 200-nanometer diameter. The phase-change memory architecture is characterized by a strip of chalcogenide material lying on top of a TiN heater element, with a thickness of five nanometers and a width of 100 nanometers. The chalcogenide layer is a germanium-antimony-tellurium alloy deposited by sputtering deposition and is 50-nanometer thick. A fifth layer of metal is deposited on top of both phase-change memories and filamentary memristors.

Two different integrated circuits were used in this article, one with filamentary-based memristors and the other phase-change memories (Fig. 2). In both architectures, each memory cell is accessed by a transistor, giving rise to a one-transistor-one-resistor (1T1R) unit cell. The transistor, used as a selector, was essential to control the programming current allowing multi-level programming of filamentary memristors. The phase-change memory chip was an array of 16,384 1T1R structures, only individually accessible. The filamentary-based memristor chip was an array of 1,024 1T1R cells arranged in a 32 × 32 configuration. This array enabled the selection of multiple memory points capable of performing parallel multiply and accumulate operations. Digital drivers were used to select multiple cells in parallel controlling the word lines (WLs), source lines (SLs), and bit lines (BLs). This array is described in detail in Supplementary Note 13.

### Iterative programming

The iterative programming methods adopted for filamentary-based and phase-change memories are different. For filamentary memristors, each device is re-programmed multiple times, with the same conditions, until its conductance reaches the target value (Algorithm 1). For the phase-change memories the programming voltage is increased or decreased at each cycle depending on the conductance value obtained in the previous cycle (Algorithm 2).

**Algorithm 1**. Iterative programming for filamentary memristors

1: $G_{max}$ : Target conductance max
2: $G_{min}$ : Target conductance min
3: $I_{cc}$ : Compliance current for target distribution
4: $i_{max}$ : Maximum number of iteration
5: $G$ : filamentary memristor conductance
6: $G \leftarrow RESET$
7: $i \leftarrow 0$
8: **while** $i < i_{max}$:
9:      $G_0 \leftarrow SET(I_{cc})$
10:      $i \leftarrow i + 1$
11:      **if** $G_{min} < G < G_{max}$:
12:          **end**
13:      **else**:
14:          $G_0 \leftarrow RESET$
15: **end**

**Algorithm 2**. Iterative programming for phase change memories

1: $G_{max}$ : Target conductance max
2: $G_{min}$ : Target conductance min
3: $V_s$ : Applied voltage
4: $V_{max}$ : Maximum voltage
5: $\delta V$ : Voltage increment
6: $G$ : phase change memory conductance

7: $G \leftarrow RESET$
8: $V_s \leftarrow V_{init}$
9: **while** $V_s < V_{max}$ and $G < G_{min}$:
10:      $G \leftarrow SET(V_s)$
11:      $Vs \leftarrow Vs + \delta V$
12: **while** $V_s < V_{max}$ and $G > G_{max}$:
13:      $G \leftarrow RESET(V_s)$
14:      $Vs \leftarrow Vs - \delta V$
15: **end**

Before the filamentary based memristors chip can be used, it is necessary to form all the devices. The forming operation consist in the following conditions: $V_{sl} = 0$ V, $V_{wl} = 1.6$ V, $V_{bl} \in [1.6, 4]$ V. The standard SET conditions are as follows: $V_{sl} = 0$ V, $V_{wl} \in [1.4, 2.2]$ V, $V_{bl} = 1.8$ V. The standard RESET conditions used are as follows: $V_{sl} = 2.6$ V, $V_{wl} = 4.8$ V, $V_{bl} = 0$ V. The off-chip generated voltage programming pulses have a pulse width of 1 $\mu$s for the SET and 100 ns for the RESET.

For the phase change memory chip, the standard SET conditions are as follows: $V_{sl} = 0$ V, $V_{wl} \in [2, 3]$ V and $V_{bl} = 4$ V. The standard RESET conditions used are as follows: $V_{sl} = 0$ V, $V_{wl} \in [0.9, 4]$ V and $V_{bl} = 4.8$ V. The off-chip generated voltage programming pulses have a pulse width of 300 ns and a rise time of 20 ns. The fall time is 1500 ns for the SET and 20 ns for the RESET.

### Correspondence between weight and conductance

The mapping between the mean and standard deviation of the normal distributions obtained after software training, $\theta_s = (\mu_s, \sigma_s)$, and the corresponding experimental conductance distributions $\theta_e = (\mu_e, \sigma_e)$, expressed in micro siemens, is a critical step. The normal distributions for weights chosen by the training algorithm and reported in Fig. 4 are mapped to conductance values in micro siemens using a scaling factor $\gamma$

$$\mu_e = \gamma \cdot \mu_s \qquad \sigma_e = \gamma \cdot \sigma_s. \tag{3}$$

We obtained the value of this scaling factor by performing a grid search to minimize the Kullback-Leibler divergence between the experimental and simulated normal distributions:

$$\gamma = \underset{\gamma \in \mathbb{R}}{\arg\min} \sum_{j \in [1,S]} \min_{i \in [1,E]} KL(\theta_{e_i}, \gamma \times \theta_{s_j}), \tag{4}$$

where $S$ is the number of the software normal distributions and $E$ is the number of available experimental normal distributions. This operation was performed at the end of the training process.

### Training using Bayes by Backprop and technological loss

The training of a Bayesian neural network consists of computing the most likely models (i.e. the posterior distribution, $p(\Omega|D)$) underlying the training dataset, $D$, and the prior belief, $p(\Omega)$:

$$p(\Omega|D) = \frac{p(D|\Omega)p(\Omega)}{p(D)}. \tag{5}$$

Here $\Omega$ represents the neural network parameters, $p(D|\Omega)$ is the likelihood, and $p(D)$ is the evidence. This equation is unfortunately intractable. Variational Inference approximates the posterior distribution, $p(\Omega|D)$, with a simpler variational distribution, $q(\Omega|\theta)$, which structure is easier to evaluate[44]. Typically the variational distributions are normal distributions, where the variational parameters $\theta = (\mu, \sigma)$ represent their mean and standard deviation. The approximation of the $\theta$ parameters, $\theta^*$, are calculated minimizing the Kullback-Leibler (KL) divergence between the variational distribution, $q(\Omega|\theta)$, and the posterior, $p(\Omega|D)$, as shown in Eq. (6). The KL divergence is a measure of the similarity between the two distributions. The calculation of the $\theta^* \in \mathbb{R}$ parameters is achieved by backpropagation[19]. This combination of variational inference and backpropagation is called Bayes by

Backprop and has been proved to be efficient for complex applications[15]. It identifies

$$\boldsymbol{\theta}^* = \underset{\boldsymbol{\theta} \in \mathbb{R}}{\arg\min}(KL[q(\boldsymbol{\Omega}|\boldsymbol{\theta})||p(\boldsymbol{\Omega}|\boldsymbol{D})])). \tag{6}$$

As illustrated in Fig. 4, resistive memories cannot implement all possible normal distributions, no matter the chosen technology flavor. The search of $\boldsymbol{\theta}^*$ should be limited inside Γ, where Γ represents the ensemble of experimental normal distributions that can be built with a given technology. To impose that $\boldsymbol{\theta}$ belongs to Γ, a "technological loss" term has been added to Eq. (6). The "technological loss" term is defined as $-\log(U_\Gamma(\boldsymbol{\theta}))$ and Eq. (6) becomes:

$$\begin{aligned}\boldsymbol{\theta}^* &= \underset{\boldsymbol{\theta} \in \Gamma}{\arg\min}(KL[q(\boldsymbol{\Omega}|\boldsymbol{\theta})||p(\boldsymbol{\Omega}|D)]) \\ &= \underset{\boldsymbol{\theta} \in \mathbb{R}}{\arg\min}\big(KL[q(\boldsymbol{\Omega}|\boldsymbol{\theta})||p(\boldsymbol{\Omega}|D)] - \log(U_\Gamma(\boldsymbol{\theta}))\big).\end{aligned} \tag{7}$$

We construct the function $U_\Gamma$, using the experimental data points of Fig. 3e, with the aim of approximating a uniform function over Γ, while ensuring differentiability at the boundary

$$U_\Gamma(\theta) = \tanh(\beta f(\theta)), \tag{8}$$

with $f(\theta)$ defined as

$$f(\theta) = \frac{1}{\delta\sqrt{2\pi}}e^{-\frac{(\theta-\theta_{exp})^2}{2\delta^2}}, \tag{9}$$

where $\theta = (\mu, \sigma)$ and $\theta_{exp}$ is the nearest experimental point to $\theta$ achieved in hardware (i.e., presented in Fig. 3e). The parameters $\delta$ and $\beta$ control the rate of increase of the technological loss outside Γ and the speed at which the technological loss approaches a minimum value close to the experimental points, respectively. In our training, we selected the values $\delta = 0.1$ and $\beta = 10$. When the value of $\theta = (\mu, \sigma)$ is significantly different from the closest experimental value $\theta_{exp}$, $f(\theta)$ approaches zero, resulting in a large value for the technological loss. This has the effect of penalizing such values of $\theta$ and encourages the network to decrease the overall loss by bringing $\theta$ closer to $\theta_{exp}$. Conversely, when $\theta$ is sufficiently close to $\theta_{exp}$, $f(\theta)$ is large enough to cause $U_\Gamma$ to saturate to 1, thereby resulting in a null technological loss. In such cases, the network is rewarded for such values of $\theta$ and can reduce the overall loss according to the standard rules of Bayes By Backprop. This balance between the use of Bayes By Backprop and the incorporation of the technological loss is achieved in a continuous and smooth manner, ensuring that the network can effectively learn and optimize while taking into account the technological constraints. Note that the experimental points $\theta_{exp}$ correspond to measurements of the devices, but six seconds after programming, in the case of filamentary memristors, and one day after programming, in the case of phase change memories. This choice ensures the stability of the programmed Bayesian neural network (see Supplementary Notes 4, 5, and 6).

The cost function resulting from Eq. (7) can be denoted as:

$$F(\boldsymbol{D}, \boldsymbol{\theta}) = KL[q(\boldsymbol{\Omega}|\boldsymbol{\theta})||p(\boldsymbol{\Omega})] - \mathbb{E}_{q(\boldsymbol{\Omega}|\boldsymbol{\theta})}[\log(p(\boldsymbol{D}|\boldsymbol{\Omega}))] - \log(U_\Gamma(\boldsymbol{\theta})) \tag{10}$$

This cost function can be minimized with classical Bayes by Backprop, and it ensures that the $\boldsymbol{\theta}^* \in \Gamma$.

To compare the computational effort of the proposed hardware-calibrated training with the classical Bayes by Backprop method, we conducted a series of experiments to measure the time required for 100 epochs on a batch of 100 images using each approach. Our findings indicate that the network trained with the classical Bayes by Backprop method completed the training process in 22 seconds, while the hardware-calibrated training algorithm took 120 seconds. This implies that our model's training process is approximately six times more computationally demanding than the classical Bayes by Backprop approach. However, it is crucial to emphasize that this training phase is a one-time requirement for the model. Once the model is trained, it can be deployed on any chip for inference at the edge.

## Mapping synaptic weights to memory arrays

After the training process, synaptic weights can be programmed to the memory arrays. The training process gives, for each synapse a mean weight value $\mu$ and a weight standard deviation $\sigma$. To convert these Gaussian distribution parameters into microsiemens, the scaling factor, $\gamma$, is utilized (see section Correspondence between weight and conductance). Second, each Gaussian is associated with the closest experimental data point obtained by programming two memory cells (Figure 4e). The metric used for this association is the Kullback-Leibler divergence. Third, each Gaussian is transferred onto $M$ crossbar arrays, where $M$ corresponds to the number of samples of a Bayesian probabilistic weight. It is important to note that each sample corresponds to two memory cells, representing one positive and one negative weight.

## Experimental setup for arrhythmia classification

The considered Bayesian neural network featured 32 inputs, 16 hidden neurons in the first layer and nine output neurons, corresponding to the nine different diseases (classes), in the second layer. Since we use conductance subtraction between two filamentary memristors to store one weight, our $32 \times 32$ crossbar array could take 32 inputs and produce 16 outputs. To realize one sample of our two-layer neural network one and a half crossbar arrays are required ($32 \times 32$ cells for the first layer and $16 \times 18$ cells for the second one). A Bayesian neural network is the collection of several ($M$) samples, so when using $32 \times 32$ crossbar arrays, $1.5M$ arrays are needed. We fully characterized 15 crossbar arrays to implement a Bayesian neural network with $M = 10$ samples. To reproduce a Bayesian neural network with more than $M = 10$ samples, we recycled the 15 crossbar arrays exploiting the fact that the cycle-to-cycle and device-to-device variability are similar in filamentary memristors[26] (see Supplementary Note 3). Therefore, by reprogramming the 15 arrays 5 times, which is equivalent to using 75 arrays, we obtain a Bayesian Neural Network with $M = 50$ samples. The arrays were programmed using the mapping technique described in the previous section.

The input data are ECG recordings[37]. A single heartbeat is a 700 ms recording, and it is converted into 32 features through a Fast-Fourier Transform (FFT). The 32 extracted features are the input of the $M$ samples of the Bayesian neural network. Since the digital drivers generate only a single read voltage level, $V_{read}$ (see Supplementary Note 13), each feature is converted into three-bit binary values, ($X_j$ with $j = 0, ..., 2$). The three bits are applied sequentially to the input of the first layer of each $s$ sample, with $s = 1, ..., M$. Each input voltage vector $\boldsymbol{X_j}$ is applied on the bit lines of the $32 \times 32$ crossbar array to generate output current vector ($V_{read}$, is applied to the selected bit lines, which correspond to an input one, the unselected bit lines are floating, which correspond to an input zero). The measured output current at the source lines, is the dot product operation through the first layer, $\boldsymbol{W_s} \cdot \boldsymbol{X_j}$, where $\boldsymbol{W_s}$ are the conductance values of a given sample (model realization) $s$. The output current for a given three-bit binary input is

$$I_{i,s} = \frac{\boldsymbol{W_s} \cdot \boldsymbol{X_0} + 2 \times \boldsymbol{W_s} \cdot \boldsymbol{X_1} + 4 \times \boldsymbol{W_s} \cdot \boldsymbol{X_2}}{7}. \tag{11}$$

Using these experimental values, we calculate the activation functions of the hidden neurons

$$a_{i,s} = \frac{I_{i,s}^+ - I_{i,s}^-}{\gamma \cdot V_{read}}, \tag{12}$$

where $\gamma$ is the scaling factor calculated with Eq. (4). Each activation function is converted to three-bit binary values. This operation is equivalent to the calculation of a clipped rectified linear unit (ReLu) activation function. The same method is applied to the second layer, in which the calculated activations are the new input. The probability that the input data $X$ belongs to a given output class $c$ for a given sample $W_s$ using a softmax function is

$$p(y = c|X, W_s) = \frac{e^{a_{c,s}}}{\sum_{j=1}^{N} e^{(a_{j,s})}}. \quad (13)$$

The disease classification (i.e., the probability that the input data belong to a specific class of disease) is the average of the probability values calculated with Eq. (13) over the number of samples. The predicted class is calculated as the argmax of the disease classifications. The aleatoric and epistemic uncertainty are calculated with Eqs. (14), (15) and (16).

### Uncertainty calculation

Unlike conventional artificial neural networks, where the output values for predictions are point estimates, Bayesian neural networks provide predictive distributions. The total uncertainty in the prediction, i.e., the predictive uncertainty, can be calculated based on the softmax of the predictive distributions calculated according to Eq. (13):

$$U_p = -\sum_{c=1}^{N} \left(\frac{1}{M}\sum_{s=1}^{M} p(y = c|X, W_s)\right) \log\left(\frac{1}{M}\sum_{s=1}^{M} p(y = c|X, W_s)\right). \quad (14)$$

The predictive uncertainty (Eq.(14)) is the sum of epistemic and aleatoric uncertainties

$$U_p = U_a + U_e. \quad (15)$$

Decomposing the predictive uncertainty is important, as epistemic and aleatoric uncertainties give us different information. High epistemic uncertainty suggests that the input data is an outlier relative to the training data set. More training data near can therefore reduce epistemic uncertainty, but does not help aleatoric uncertainty. Aleatoric uncertainty is uncertainty in data, to reduce it more refined input data are required (e.g., more powerful sensors). The aleatoric uncertainty can be obtained as:

$$U_a = -\frac{1}{M}\sum_{s=1}^{M}\sum_{c=1}^{N} p(y = c|X, W_s) \log p(y = c|X, W_s). \quad (16)$$

### Inference energy consumption estimates

To estimate the energy consumption of the Bayesian neural network we first calculated the number of dot product operations for one inference:

$$Operations = 4 \cdot I_l \cdot H_l + 4 \cdot H_l \cdot O_l. \quad (17)$$

Here $I_l$ is the input length, $H_l$ is the hidden layer length, and $O_l$ is the output length. The factor four is due to fact that each sample of a Bayesian probabilistic weight is stored as the difference between the conductance values stored in two memory cells and that a dot product contains addition and multiplication. One inference costs 2,624 operations. The cost of a single analog Multiply-and-Accumulate (MAC) operation in a resistive memory-based analog in-memory computing circuit depends on the input and output size and on the weight precision, and can vary considerably depending on the memory technology, CMOS node, array size, and design choices. We relied on energy per operation number of three industrial platforms employing resistive memory[3], magnetoelectric memory (MRAM)[39], and phase

change memory[7]. The results are reported in Supplementary Note 15. We found a cost ranging between 0.7 and 2.5 nanojoules per inference. Note that these estimates consider only the Multiply-and-Accumulate, which we expect to dominate. Still, additional circuitry will be needed, e.g., to present the input, analyze the outputs, and transfer data between arrays of the neural network.

To gain a perspective on the energy efficiency of the proposed approach compared to conventional hardware, we benchmarked this figure to the energy required for running the operations to perform inference of the same Bayesian neural network on an STM32F746ZGT6 MCU (integrated on a test Nucleo-F746ZG board), which is typically used for edge AI applications. These operations coded in the C language using the ST Microelectronics STM32 Cube integrated development environment and compiled and built without debugging options and using the strongest optimizations for speed (-Ofast option). To provide a fair comparison with our in-memory-computing platform, our C code includes only the multiply-and-accumulate operation. (We controlled that multiply-and-accumulate operations represented more than 99% of the execution time of our program.) We timed our program, and measured the current consumption of the MCU using an Ampere meter (we measured the current solely consumed by the MCU, excluding any other component of the board). The STM32 MCU is fabricated in a 90-nanometer CMOS node. We found a consumption of 170 microjoules per inference (with ten samples of the Bayesian neural network).

We have included another benchmark using an NVIDIA Jetson Nano, an edge-computing board widely used for edge AI applications equipped with an NVIDIA Tegra X1 system on chip, featuring a GPU and a multicore CPU. This chip-based system is manufactured in a more modern 20-nanometer CMOS node. In our benchmarch test, we perform the multiply-and-accumulate operations of our system on the GPU. Our benchmark code is written using Pytorch 1.10 with NVIDIA Jetpack 4.6 and NVIDIA CUDA 10.2. All the multiply-and-accumulate operations for the different output samples are performed with a single tensor multiplication using the Pytorch torch.matmul function, ensuring a fully parallel operation and an optimal use of the GPU. Additionnally, our code allows batching, i.e., the processing of several inputs simultaneously within the same torch.matmul call. As described in the Discussion section of the paper, higher batching allows a better use of the resources of the GPU and reduces the energy consumption per input. To obtain a reliable estimate of energy consumption, with repeated the multiply-and-accumulate operations multiple times and timed the process using the repeat function from the Python 1.10 timeit library. We chose the number of repetitions to reach a total computation time of a minute, allowing the power consumption of the board to stabilize. During these measurements, we monitored the power consumption of both the NVIDIA Jetson Nano GPU and the whole system, using the built-in power monitoring feature of the board. The energy consumption is obtained by multipliying the computation time of a torch.matmul call by the power consumption.

## Data availability

All the measured data are available upon request.

## Code availability

All software programs used in the presentation of the Article are available upon request.

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

## Acknowledgements

This work was supported by European Research Council consolidator grant DIVERSE (reference: 101043854) and by European Research Council starting grant NANOINFER (reference: 715872). It also benefits from a France 2030 government grant managed by the French National Research Agency (ANR-22-PEEL-0010). In addition, we thank L. Hutin, S. Bonnetier, F. Andrieu, J. Arcamone, J. Grollier, P. Bessière and J. Droulez for discussing various aspects of the article.

## Author contributions

D.B. and T.H. proposed the initial idea of the hardware-calibrated training algorithm. D.B, T.H. D.Q, and E.V. conceived the experiments. D.B. and V.M. performed the experiments with the phase-change memory array. D.B., S.M., and N.C. performed the inference measurements on the two-layer Bayesian neural network. D.B. and T.H. conducted the software experiments and analysed the data. T.D. and A.M. performed preliminary studies concerning Bayesian neural networks and uncertainty evaluation. E.E. designed the circuits, under the supervision of J.M.P. The circuits were fabricated at CEA-Leti under the supervision of J.F.N. and G.B. D.Q. and E.V. supervised the work and wrote the initial version of the manuscript. All authors discussed the results and reviewed the manuscript.

## Competing interests

The authors declare no competing interests.
