## [Peer Review File · Nature Communications]

Review on the manuscript “**Bringing uncertainty quantification to the extreme-edge with memristor-based Bayesian neural networks**” authored by Djohan Bonnet *et al.*, submitted to *Nature Communications*. [Manuscript Number: NCOMMS-23-00214]

In this manuscript, the authors reported a memristor-based Bayesian neural network (BNN) using the stochasticity of memristors. In this study, two memristors were used to solve the problem of correlation between standard deviation and mean. The memristor-based BNN was trained using arrhythmia ECG data, and two different types of uncertainty were quantified. Using epistemic uncertainty, unknown data was almost perfectly distinguished from trained data. The performance of the fabricated memristor-based $75 \times 32 \times 32$ BNN hardware (Figure 5) was impressive. This reviewer basically agrees with the publication of this work in this journal. However, the present form of the manuscript needs substantial reorganization because it is difficult to understand the work's crucial point. The detailed comments are the following:

1. First, it would be much better to explain how the BNN in software works briefly in the introduction. The current manuscript explains the limitation of conventional (or deterministic) neural networks for an outlier, and BNN performs better in identifying such a problem. However, it does not explain how the software BNN works during the training and inference steps. This explanation is especially necessary to understand why the μ and σ values must vary independently over a space as large as possible. This will also help the readers understand the limitations of the current memristors and PCM devices regarding the partial decorrelation of the two parameters.
2. As the authors already mentioned, using two memory cells per weight gives better μ vs. σ performance. It would be better to explain a bit more why this was the case.
3. Related comment 2: one of the most crucial factors for implementing the memristor-based BNN is solving the problem of the correlation between μ and σ using the device physics. In this study, the method of using two memristors proposed in previous studies was used to solve the correlation problem. This reviewer believes that technological loss is to compensate for insufficient decorrelation. Therefore, the “technological loss” term appears to be the most crucial factor in determining the novelty of this paper, but there is not enough explanation for the technological loss in this manuscript. A detailed description of how the technological loss was defined and why it is defined as such needs to be added. In addition, it is necessary to explain the principle of how the technological loss matches the classical variational inference domain (θ) to the domain representable in the memristor (Γ). Also, there was almost no explanation of how this technological loss term was used during the training step, making a clear understanding of this important term difficult.

4. A related comment is that the explanation for Fig. 4 is too terse, which must comprise the most crucial part of this work. To this reviewer, the scattered data in (a) seem to be the weight (or μ and σ) values calculated off-chip during the training step without considering the technological loss term. In (b), these values are corrected to fit into the practically available μ and σ range (inside the solid line), but how this could be the case was not explained in detail, despite its importance.
5. In this work, the random statistical variations of the weight deviations in the software BNN during each training and inference step were replaced (represented) by the random variations of the memristor and PCM devices of the 75 CBAs, which represent the cell-to-cell variation. However, it is unclear if the cell-to-cell variation of the memristors and PCM, shown in Fig. 3, can replace the step-to-step variation in the software BNN. A related comment is that each of these memory cells has switching cycle-to-cycle variations. Could this additional variation have any influence on the performance?
6. Figure 1 should be expressed more clearly. For example, in Fig 1a and 1d, the number of output neurons and the output current should be displayed accurately according to the number of classification classes. (or insert an omission mark) This reviewer initially had a problem understanding this figure because there are three classification cases (clear, unclear, unknown) and three output neurons, which invoked a feeling that each output neuron represents the three classification cases. Finally, it was understood that these output neurons still correspond to the 10 classes of the ECG signals.
7. Until Figure 4, filamentary memristor and PCRAM were explained together, but Figure 5 shows only the result of the filamentary memristor, which is confusing. Adding the PCRAM classification results to Fig. 5 or moving the PCRAM contents of Figs 2 to 4 to the supporting information would improve the consistency of the work.
8. A more detailed explanation of the array mapping method needs to be added. In addition, it is necessary to analyze how much the performance of BNN depends on the distribution when the distribution is adjusted using the program-and-verify method. Was the result vary significantly depending on the programming method?
9. An analysis of why memristor-based BNN outperforms the ideal case (float 32) should be added. It is not well understood that the performance of memristor-based BNN, where the correlation of statistical factors due to device physics still exists, is better than the ideal case free from the problem.
10. According to the inference process flow shown in Supplementary Figure 6 and the main text, the device conductance data should be trained off-chip by the

backpropagation algorithm. Does the power mentioned in the main text include this neural network training process?

11. Minor comments: There are typos and grammar errors.

- The caption of Fig. 2 also needs to be revised. Figures 2 (a), (b), (d) and (e) are TEM images, not SEM images.

- In Fig. 5c on page 5, 'exprimental' → 'experimental', above the graph.

- In line 172 on page 7, 'plotted in black' → 'plotted in red'

- In supplementary Fig.6, 'To each distribution correspond 2 programming conditions~~' → 'Each distribution corresponds to 2 programming conditions~~', 'The three currents give an activation', and 'The activation gives three input patterns~~'

Reviewer #2 (Remarks to the Author):

This manuscript reports the implementation of Bayesian neural network using memristor-based electronics, a critical improvement toward efficiently implementing Bayesian inference based on the authors' previous work [1]. As a demonstration, such electronics is used to process classification of heartbeats and estimate the certainty of predictions. This approach has advantage of lower energy consumption comparing to conventional CMOS electronic implementations.

[1] Dalgaty, T., Esmanhotto, E., Castellani, N., Querlioz, D. & Vianello, E. Ex Situ Transfer of Bayesian Neural Networks to Resistive Memory-Based Inference Hardware. *Advanced Intelligent Systems* 3, 2000103 (2021).

My comments are listed below:

1.This is not the first paper about using memristors to implement Bayesian neural network. Previous papers [2][3] and the authors' previous work [4] proposed to exploit different sources of variability in memristors to implement Bayesian inference. It is important to cite them in the state of the art and illustrate the novelty of this paper, if possible, compare with them.

[2] Lin, Y. et al. Bayesian Neural Network Realization by Exploiting Inherent Stochastic Characteristics of Analog RRAM. in 2019 IEEE International Electron Devices Meeting (IEDM) 14.6.1-14.6.4 (IEEE, 2019). doi:10.1109/IEDM19573.2019.8993616.

[3] Li, X. et al. Enabling High-Quality Uncertainty Quantification in a PIM Designed for Bayesian Neural Network. in 2022 IEEE International Symposium on High-Performance Computer Architecture (HPCA) 1043–1055 (IEEE, 2022). doi:10.1109/HPCA53966.2022.00080.

[4] Dalgaty, T., Vianello, E. & Querlioz, D. Harnessing intrinsic memristor randomness with Bayesian neural networks. in 2021 International Conference on IC Design and Technology (ICICDT) 1–4 (IEEE, 2021). doi:10.1109/ICICDT51558.2021.9626535.

2.The novelty of the research is not strong, in other words, not displayed. For example, the first innovation of this work, that is, training Bayesian neural network using variational inference, has already been reported for several times [2, 3]. And authors should provide some comment on the computational efforts/cost of the off-chip training phase.

3.In the manuscript, when implementing the network, authors sample M weight values for each synapse based on its probability distribution, and program them to the M memory array. I think it is using 2xM memristors to the store sample value of a weight. However, the authors claimed that using 2 memristors to implement each probability synapse weight. Please clarify this.

4.The γ scaling factor and the technological loss term both allow hardware to implement the weight using memristor. What will the network performance be like with or without the technological loss term? And it would be better to add some explanations about the definition of the γ scaling factor (Eq.4).

5.For germanium-antimony-tellurium phase-change memories, it would be better to study how the network performance is after one hour, one day, and two weeks. For hafnium-oxide-based filamentary memristors, iterative programming techniques suffer from conductance relaxation [5], which could also cause conductance drift like phase-change memory devices. Also, device-to-device variability could also induce weight deviation. Their impacts on network performance should also be investigated. It would be good if the authors could provide single-weight plots of target and measured distribution.

[5] Wan, W. et al. A compute-in-memory chip based on resistive random-access memory. Nature 608, 504–512 (2022).

6.The demonstrated network models (32x16x9) and the dataset (ECG data) are quite simple. In order to show the effectiveness of the proposed method more convincing, more complex simulated or experimental network models and datasets (e.g., MNIST/CIFAR10) are required.

7.The manuscript lacks benchmark and comparison with other in-memory circuit implementation, such as [1][2][3][4], which could significantly improve the impact of this work.

8.It would be better if the authors could improve the organization of this paper, especially introduction section to highlight the novelty. It would be good if the authors could strengthen that energy consumption estimation section.

9.Minor comments:

In line 172 : “while the unseen disease data points are plotted in black” might be “while the unseen disease data points are plotted in red”.

Reviewer #3 (Remarks to the Author):

The authors proposed a hardware-based Bayesian neural network (BNN) by employing the intrinsic random resistance distribution of memristors (filamentary oxide memristor or phase change memory) for edge applications, such as medical diagnoses. Some specific designs are reported: (1) variational inference; (2) the synapse is evaluated using two memristors; (3) A critical “technological loss” is introduced. In the application of classifying heartbeats signals, the proposed BNN shows comparable accuracy, aleatoric and epistemic uncertainty estimations to software simulations.

My main concern about this manuscript is the novelty. The same team has already reported several studies combining Bayesian theory with memristors. For example, in the work (“A memristor-based Bayesian machine, Nature Electronics, 6, 52–63, 2023”), the authors used a memristor array to implement the Bayesian inference where the probability is encoded as a bit stream. In this manuscript, the authors describe the difference between this work and the prior work as “Ref.18 exploited the probabilistic nature of memristors to perform Bayesian learning. This approach can only be applied to small-scale tasks, but unlike Bayesian neural networks, it does not suffer from the limitations imposed by the correlation of mean value and standard deviation of memristors”. However, I notice that there are substantial similarities as follow:

- (1) Both are dedicated to energy-efficient edge applications
- (2) Both works focus on using the random resistance of memristor to implement the Bayesian inference/reasoning
- (3) Both have similar architecture
- (4) Same programming method: compare or add/subtract two memristor units to get random values or conductance

In my opinion, this manuscript is an incremental improvement of the previous work, and the improvement is insufficient for a publication in the Nature Communications.

Other Issues:

☐A major issue of this work is related to the number of neural networks and synaptic weight, which causes a high energy consumption: 270 nJ/inference. It is better than the conventional GPU hardware, but not competitive with memristive neural networks. For edge applications, power consumption is a key concern. Can the authors demonstrate how to overcome this shortcoming, and balance the necessity of uncertainty prediction and the power consumption? Apart from BNNs, a lot of work has been done to predict uncertainty in ANNs. Besides BNN, is it possible to use memristors to implement a high energy-efficient ANN with uncertainty prediction?

☒The authors argue that the mean value and the standard deviation of the resistance are correlated for HfOx-based memristor array, while the probability distribution of parameters in BNN can “take any shape”, which is preferred. It is well reported that the resistance distribution of some oxide memristors is irregular. Why not use a memristor array that shows a random resistance distribution, which favors the formation of BNN? What are the consideration when define the structures?

☒The domains of normal distribution of filamentary memristor and phase-change memory are quite different. How does this distribution affect the accuracy, aleatoric and epistemic uncertainty estimations? It is stated that “To extend the domain of normal distributions, we store each sample of a probabilistic weight as the difference between the conductance values of two adjacent memory cells”. Can we further extend the domain of normal distributions by using more memory cells? What are the pros and cons? If one memory cell is used, do the accuracy, aleatoric and epistemic uncertainty estimations deteriorate?

☒BNNs provide a high prediction confidence, but the classification accuracy is around 75%, which is much lower than other reported memristor-based neural network for the similar bio-signal classification (~95%, *Sci. Adv.* 2020; 6: eabc4797; *Nat Commun* 11, 4234, 2020). What is the reason for the poor accuracy? Can it be improved to a comparable level?

☒The performance of the memristive neural network should be added to Table 1 to make the comparison more convincing. The authors claim 800 times improvement in energy efficiency comparing to the conventional GPU platform. However, the benchmarks and the comparison method for GPU and memristor BNN are unclear. More details should be provided. It will make more sense to compare at a system level rather than focusing on the processing part.

Overall, I do not recommend publishing this in Nature Communications, based on the limited novelty. The manuscript is more suitable for other journals, such as Scientific Reports.

Response to Reviews

We would like to thank the anonymous reviewers for their time and excellent feedback, which has allowed us to improve the quality of our manuscript. We have addressed the points raised by the reviewers and revised the manuscript accordingly. The revision includes multiple new results and analyses, presented over 11 new Supplementary notes, 2 new Supplementary Tables, and 10 new Supplementary Figures. We have also reworked the article for clarity, better benchmarking, and better positioning with regard to the state of the art.

Reviewer 1 (Remarks to the Author)

General comments

In this manuscript, the authors reported a memristor-based Bayesian neural network (BNN) using the stochasticity of memristors. In this study, two memristors were used to solve the problem of correlation between standard deviation and mean. The memristor based BNN was trained using arrhythmia ECG data, and two different types of uncertainty were quantified. Using epistemic uncertainty, unknown data was almost perfectly distinguished from trained data. The performance of the fabricated memristor based $75 \times 32 \times 32$ BNN hardware (Figure 5) was impressive. This reviewer basically agrees with the publication of this work in this journal. However, the present form of the manuscript needs substantial reorganization because it is difficult to understand the work's crucial point

We thank the reviewer for his/her review and these comments.

Comment 1:

First, it would be much better to explain how the BNN in software works briefly in the introduction. The current manuscript explains the limitation of conventional (or deterministic) neural networks for an outlier, and BNN performs better in identifying such a problem. However, it does not explain how the software BNN works during the training and inference steps. This explanation is especially necessary to understand why the μ and σ values must vary independently over a space as large as possible. This will also help the readers understand the limitations of the current memristors and PCM devices regarding the partial decorrelation of the two parameters.

Based on this comment and similar ones, we have entirely rewritten the introduction of our article. We reproduce here the part of the new introduction addressing this comment:

The intrinsic randomness of memory nanodevices aligns naturally with the random variable nature of synapses in Bayesian neural networks. An implementation of Bayesian neural networks with memory nanodevices can be achieved by programming a neural network M times to reproduce the sampling operation necessary to derive M conventional neural networks from the Bayesian one. However, a critical question remains: how can we train Bayesian neural networks to align with the characteristics of memory nanodevices?

Synaptic weight probability distributions in Bayesian neural networks can take any shape^{1,2}, but the statistical properties of memristors and phase change memories follow rigid physics rules³⁻⁵. Filamentary memristors, for instance, demonstrate broader probability distributions at higher resistance states and narrower ones at lower resistance states^{6,7}, correlating resistance mean value and standard deviation. To overcome this difficulty, two recent studies proposed new devices with tunable inherent resistance probability distributions, using two-dimensional materials⁸ and magnetic devices⁹. These solutions, were validated with simulations of Bayesian neural networks.

In the main contribution of our paper, we propose a dedicated technique for Bayesian neural networks – variational inference augmented by a “technological loss” that leads to networks readily implementable with more conventional memory nanodevices. Standard variational inference trains the mean value and standard deviation of each synapse via backpropagation to identify plausible interpretations of the training data.

During the training process, the mean value and standard deviation of a synaptic weight evolve following different gradient values and become fully decorrelated. Our added technological loss constrains these synaptic weights and standard deviations to domains implementable with the selected nanodevices for a given technological implementation. We demonstrated this technique's effectiveness using standard nanodevices (filamentary memristors) to accomplish the first complete nanodevice-based Bayesian neural network implementation for a real-world task—classifying types of arrhythmia recordings with precise

aleatoric and epistemic uncertainty. Our system utilized 75 arrays of fabricated 32×32 memristor chips integrating hafnium oxide memristors and CMOS peripheral circuitry for in-memory computations based on Kirchoff's laws.

Comment 2:

As the authors already mentioned, using two memory cells per weight gives a better μ vs σ performance. It would be better to explain a bit more why this was the case.

In Figure 1a, we present the domain of normal distributions obtained using one (blue) and two (green) memory cells per sample of a Bayesian probabilistic weight. When employing only one cell per sample, the domain of normal distributions is limited to two lines, representing the distributions obtained with and without iterative programming, along with an additional point representing the reset operation. Using two cells per sample allows for a reduction in the dependencies between μ and σ , while simultaneously increasing the available space for both parameters (μ and σ).

As demonstrated in Figure 1b, using two cells per sample significantly improves network performance compared to the case of one cell per sample. A broad spectrum of possibilities for μ is essential for achieving high accuracy, while a broad spectrum of possibilities for σ is necessary for uncertainty estimation. Synapses with small σ values ensure minimal output variability for well-known situations. Conversely, synapses with large σ provide output variability for unknown input data, facilitating uncertainty estimation.

In the revised version of the manuscript, these new results are included in the new Supplementary Note 7.

Figure 1. Impact of the number of memristors per sample of a Bayesian probabilistic weight **a** Domain of the normal distribution obtained experimentally by storing samples on one (blue), and two (green) memristors. μ and σ are normalized to the minimum achievable standard deviation, σ_{min} , obtained with different experimental conditions. **b** Network performances obtained by storing samples on one, and two memristors. The training process was repeated ten times, and each inference was repeated 50 times with $M = 50$ samples. The final result represents the average performance obtained from the 500 inferences.

Comment 3:

Related comment 2: one of the most crucial factors for implementing the memristor-based BNN is solving the problem of the correlation between μ and σ using the device physics. In this study, the method of using two memristors proposed in previous studies was used to solve the correlation problem. This reviewer believes that technological loss is to compensate for insufficient decorrelation. Therefore, the “technological loss” term appears to be the most crucial factor in determining the novelty of this paper, but there is not enough explanation for the technological loss in this manuscript. A detailed description of how the technological loss was defined and why it is defined as such needs to be added. In addition, it is necessary to explain the principle of how the technological loss matches the classical variational inference domain (θ) to the domain representable in the memristor (Γ). Also, there was almost no explanation of how this technological loss term was used during the training step, making a clear understanding of this important term difficult.

The proposed hardware-calibrated training is a novel variation of the popular Bayes By Backprop method. During the training process, the technological loss term, $-\log(U_{\Gamma}(\theta))$, is incorporated into the overall loss function

$$Loss = Loss_{VI} - \log(U_{\Gamma}(\theta)). \tag{1}$$

The term $Loss_{VI}$ corresponds to the standard loss used in Bayes By Backprop, while $U_{\Gamma}(\theta)$ is determined by the experimental data, θ_{exp} , with the aim of approximating a uniform function over Γ while ensuring differentiability at the boundary.

$$U_{\Gamma}(\theta) = \tanh(\beta \cdot f(\theta)), \tag{2}$$

where $f(\theta)$ is defined as

$$f(\theta) = \frac{1}{\delta\sqrt{2\pi}} e^{-\frac{(\theta - \theta_{exp})^2}{2\delta^2}}, \quad (3)$$

where $\theta = (\mu, \sigma)$ and θ_{exp} is the nearest experimental point to θ achieved in hardware (i.e., presented in Fig. 3e of the main article). The parameters δ and β control the rate of increase of the technological loss outside Γ and the speed at which the technological loss approaches a minimum value close to the experimental points, respectively. When the value of $\theta = (\mu, \sigma)$ is significantly different from the closest experimental value θ_{exp} , the function $f(\theta)$ approaches zero, resulting in a large value for the technological loss. This has the effect of penalizing such values of θ and encourages the network to decrease the overall loss by bringing θ closer to θ_{exp} . Conversely, when θ is sufficiently close to θ_{exp} , $f(\theta)$ is large enough to cause U_Γ to saturate to 1, thereby resulting in a null technological loss. In such cases, the network is rewarded for such values of θ and can reduce the overall loss according to the standard rules of Bayes By Backprop. This balance between the use of Bayes By Backprop and the incorporation of the technological loss is achieved in a continuous and smooth manner, ensuring that the network can effectively learn and optimize while taking into account the technological constraints.

We now included this information in the Methods section. The rewritten Introduction also highlights more prominently that the technological loss is the key novelty of our approach.

Comment 4:

A related comment is that the explanation for Fig. 4 is too terse, which must comprise the most crucial part of this work. To this reviewer, the scattered data in (a) seem to be the weight (or μ and σ) values calculated off-chip during the training step without considering the technological loss term. In (b), these values are corrected to fit into the practically available μ and σ range (inside the solid line), but how this could be the case was not explained in detail, despite its importance.

Figure 4a indeed presents the weight data (μ and σ) of the synapses obtained after off-chip training of our reference arrhythmia classification task, using the popular Bayes-By-Backprop method, without modification. These data are expressed in μS after the mapping factor γ has been computed, as explained in detail in the Methods section. It is evident that this weight data do not entirely fit within the range of values range achievable with filamentary memristors (materialized by the solid $\Gamma_{\text{memristor}}$ curve).

By contrast, Figure 4b shows the weight data obtained when using our new proposed hardware-calibrated off-chip training, incorporating the technological loss term in the loss function (Eq. 4), described in Comment 3. Figure 4b show that the approach is very successful: all weight data now fits within the range of values range achievable with filamentary memristors. Figures 4c and 4d show equally successful results in the PCM case.

We have now revised Figure 4 to clarify it, with clear labels and the addition of a colorbar. We have also rewritten its caption entirely and improved its description in the main text.

Figure 2. Domains of normal distribution obtained during off-chip training using the popular Bayes By Backprop method and the proposed technologically plausible off-chip training, incorporating the technological loss term in the loss function (Eq. 4). **a** Domain of normal distributions $\theta=(\mu, \sigma)$ obtained after training with the classical variational inference method and mapping the software values to the conductance range achievable with filamentary memristors (blue) and phase-change memories (green). **b** Domain of normal distributions $\theta=(\mu, \sigma)$ obtained after training with the proposed method calibrated on filamentary memristors (blue) and phase-change memory experimental data (green).

Comment 5:

In this work, the random statistical variations of the weight deviations in the software BNN during each training and inference step were replaced (represented) by the random variations of the memristor and PCM devices of the 75 CBAs, which represent the cell-to-cell variation. However, it is unclear if the cell-to-cell variation of the memristors and PCM, shown in Fig. 3, can replace the step-to-step variation in the software BNN. A related comment is that each of these memory cells has switching cycle-to-cycle variations. Could this additional variation have any influence on the performance?

This is an excellent question. Based on new measurements, we have included a thorough discussion of this point in the revised manuscript, presented in a new Supplementary Note 3. We summarize its main point here.

First, we check that device-to-device variability is stable when programming the array multiple times. (Figures 3a and 3b, for memristors and phase change memories, respectively). Each data point represents the mean and standard deviation of a Gaussian distribution obtained by programming 1,000 distinct devices, with different points representing the 1,000 cycles. We see that the cycle-to-cycle variability does not affect the mean and standard deviation of the Gaussian distribution obtained from exploiting cell-to-cell variation.

To understand this result further, Figures 3c and 3d compare the probability density of 1,000 distinct devices programmed once (device-to-device variability) with the probability density of one device cycled 1,000 times (cycle-to-cycle variability) for memristors and PCM technologies, respectively. Both sources of variability exhibit an equivalent impact on conductance variability.

The reason for the apparent equivalence between device-to-device and cycle-to-device variability differs for the two types of devices. For memristors, the conductance depends on the shape of the filament, which varies cycle-to-cycle and depends weakly on the specific properties of a particular structure. In the case phase change memories, by contrast, the equivalence

originates solely due to the use of iterative programming.

Based on these results, we do not anticipate any significant impact from the additional cycle-to-cycle variability on the network's performance.

Figure 3. Impact of cycle-to-cycle variability on cell-to-cell variations for memristors and PCMs. (Top) Scatter plot illustrating the impact of cycle-to-cycle variability on cell-to-cell variation for memristors (a) and phase change memory (PCM) (b). Each data point represents the mean and standard deviation of a Gaussian distribution obtained by programming 1,000 distinct devices, with different points representing 1,000 cycles. (Bottom) Probability density of 1,000 cells programmed under the same conditions, depicting device-to-device variability (blue), compared with the probability density of one device programmed 1,000 times for both memristor (c) and PCM (d) technologies.

Comment 6:

Figure 1 should be expressed more clearly. For example, in Fig 1a and 1d, the number of output neurons and the output current should be displayed accurately according to the number of classification classes. (or insert an omission mark) This reviewer initially had a problem understanding this figure because there are three classification cases (clear, unclear, unknown) and three output neurons, which invoked a feeling that each output neuron represents the three classification cases. Finally, it was understood that these output neurons still correspond to the 10 classes of the ECG signals.

We thank the reviewer for this comment. We modified the Figure accordingly. The new version is reproduced in this response as Fig. 4.

Figure 4. General architecture of the Bayesian neural network. **a** Schematic of the Bayesian neural network used for heart disease (arrhythmia) classification. In Bayesian neural networks, the weights are represented by probability distributions, thus naturally including uncertainty in the model. **b** Example of output neuron activation distributions, obtained for certain output, uncertain output, due to noisy input data, and unknown data (i.e. out-of-distribution data). **c** Experimental setup. **d** Hardware implementation of a Bayesian neural network by combining multiple versions of ANNs.

Comment 7:

Until Figure 4, filamentary memristor and PCRAM were explained together, but Figure 5 shows only the result of the filamentary memristor, which is confusing. Adding the PCRAM classification results to Fig. 5 or moving the PCRAM contents of Figs 2 to 4 to the supporting information would improve the consistency of the work.

We thank the reviewer for this comment. We have followed this recommendation and included in Figure 5 of the main text the results obtained with PCM. The new version of Figure 5 in the main text is reproduced in this response as Figure 5. To ensure readability, we have also decided to move the ROC curves (Figure 6), for both PCM and OxRAM, to a new Supplementary Note 1. We have also extensively revised the discussion of Fig. 5 in the main text to adjust to these changes.

Figure 5. Measurements of the fabricated memristor-based Bayesian neural network and simulations of a PCM Bayesian neural network. **a** tSNE visualization of input data, different colors representing different classes (diseases). Nearby points correspond to similar data and distant points to dissimilar data. **b** tSNE visualization of experimental data classification. The different colors represent points correctly or incorrectly predicted and unseen data. **c** Experimental probability density distribution of the aleatoric uncertainty for correct predictions, incorrect predictions and unseen diseases, using filamentary memristors. **d** Experimental probability density distribution of the epistemic uncertainty for correct predictions, incorrect predictions and unseen diseases, using filamentary memristors. **e** Simulated probability density distribution of the aleatoric uncertainty for correct predictions, incorrect predictions and unseen diseases for a conventional neural network with the same architecture and using *float32* encoding for the synapses. **f** Simulated probability density distribution of the aleatoric uncertainty for correct predictions, incorrect predictions and unseen diseases, using PCMs. **g** Simulated probability density distribution of the epistemic uncertainty for correct predictions, incorrect predictions and unseen diseases, using PCMs. **h** Measured (memristor) and simulated (PCM) accuracy, epistemic uncertainty, and aleatoric uncertainty performance (calculated as the area of the ROC curves presented in Suppl Note 1) as a function of the number of devices per synapse.

Figure 6. Receiver Operating Characteristic (ROC). **a** ROC curve corresponding to the differentiation between correct prediction and incorrect prediction, based on aleatoric uncertainty measured on the proposed memristor-based Bayesian neural network (green) and simulated (weights stored as *float32* real numbers) for Bayesian (black) and conventional neural network with the same architecture (red). **b** ROC curve corresponding to the differentiation between known and unknown data, based on epistemic uncertainty measured on the proposed memristor-based Bayesian neural network (green) and simulated (weights stored as *float32* real numbers) for Bayesian (black) and conventional neural network with the same architecture (red). **c** ROC curve corresponding to the differentiation between correct prediction and incorrect prediction, based on aleatoric uncertainty measured on the proposed PCM-based Bayesian neural network (green) and simulated (weights stored as *float32* real numbers) for Bayesian (black) and conventional neural network with the same architecture (red). **d** ROC curve corresponding to the differentiation between known and unknown data, based on epistemic uncertainty measured on the proposed PCM-based Bayesian neural network (green) and simulated (weights stored as *float32* real numbers) for Bayesian (black) and conventional neural network with the same architecture (red).

Comment 8:

A more detailed explanation of the array mapping method needs to be added. In addition, it is necessary to analyze how much the performance of BNN depends on the distribution when the distribution is adjusted using the program-and-verify method. Was the result vary significantly depending on the programming method?

We have now incorporated a new “Mapping synaptic weights to memory arrays” section in the Methods section of the article and included a new Supplementary Note 8 analyzing the impact of the program-and-verify method. We summarize their content here.

Array mapping method: First, the desired weights, $\theta = (\mu, \sigma)$, are calculated using the proposed off-chip hardware calibrated method based on the technological loss (see the response to Comment 3). To convert the Gaussian distribution into microsiemens, the scaling factor, γ , is utilized. Second, each Gaussian is associated with the closest experimental data point obtained by programming two memory cells (Figure 3e in the main text). The metric used for this association is the KL divergence. Third, each Gaussian is transferred onto N crossbar arrays, where N corresponds to the number of samples of a Bayesian probabilistic weight. It is important to note that each sample corresponds to two memory cells, representing one positive and one negative weight.

Impact of the program-and-verify (iterative programming) method: Regarding the impact of iterative programming, Figure 7a illustrates the domain of normal distributions obtained with (green) and without (blue) iterative programming, in the case of filamentary memristors. The inclusion of iterative programming expands the range of achievable σ values in experimental results. As shown in Figure 7b, a broad spectrum of possibilities for σ improves the estimation of aleatoric uncertainty of 3% and the accuracy of 2%. Synapses with small σ ensure minimal output variability for well-known inputs. Conversely, synapses with large σ are needed to provide output variability for unknown input data.

For phase change memory, by contrast, iterative programming is fundamental to programming the devices and is used in all cases.

Figure 7. Impact of iterative programming. **a** Domain of the normal distribution obtained experimentally by storing samples without (blue) and with (green) iterative programming. μ and σ are normalized to the minimum achievable standard deviation, σ_{min} , obtained with different experimental conditions. **b** Network performances obtained by storing samples without and with iterative programming. The training process was repeated ten times, and each inference was repeated 50 times with $M = 10$ samples. The final result represents the average performance obtained from the 500 inferences.

Comment 9:

An analysis of why memristor-based BNN outperforms the ideal case (*float32*) should be added. It is not well understood that the performance of memristor based BNN, where the correlation of statistical factors due to device physics still exists, is better than the ideal case free from the problem.

We are grateful for the opportunity to clarify a point that was inadequately explained in the initial version of our manuscript. The training process of a BNN inherently possesses a random nature: when training a BNN multiple times using the same dataset, small variations arise in its performance, particularly in terms of accuracy and uncertainty evaluation. For this reason, in the initial submission, Table 1 presented the mean performance of ideal case BNNs (*float32*) across ten training processes.

For the hardware experiment, we also conducted ten training runs. Then, we selected the neural network that exhibited the best performance in software, and we programmed this network onto the 75 memristor arrays, and measured experimentally its performance, which we reported in Table 1. The side-to-side presentation of a mean result in the *float32* case, and of a chosen case in the experimental case, created the impression that the experimental outcomes slightly surpassed those of the ideal case, which is not the case.

To rectify this misinterpretation, we have now also included the “best performance” achieved in the ideal case in Table 1, which matches the experimental result in terms of accuracy performance, while slightly surpassing it in terms of raw accuracy.

	Conventional ANN (float32)	Bayesian (float32)	Bayesian Hardware (filamentary memristor experimental)	Bayesian Hardware (filamentary memristor simulation)	Bayesian Hardware (phase-change memory simulation)
Accuracy classification	best: 81% mean: 80%	best: 80% mean: 79%	75%	best: 76% mean: 76%	best: 73% mean: 73%
Prediction confidence (aleatoric) [AUC]	best: 0.90 mean: 0.79	best: 0.92 mean: 0.90	0.91	best: 0.91 mean: 0.89	best: 0.85 mean: 0.87
Anomaly detection (epistemic) [AUC]	0.5	best: 1 mean: 0.95	0.99	best: 0.96 mean: 0.92	best: 0.96 mean: 0.82

Table 1. Comparison of accuracy and uncertainty prediction performances

Changes to the manuscript: We have made several modifications to enhance the manuscript’s clarity. Table 1 now includes the best *float32* performance data. We have revised the Results section to explicitly compare the *float32* case’s performance with the experimental findings. In the Methods section, we made it clear that the BNN programmed experimentally was chosen from ten different training trials.

Comment 10:

According to the inference process flow shown in Supplementary Figure 6 and the main text, the device conductance data should be trained off-chip by the backpropagation algorithm. Does the power mentioned in the main text include this neural network training process?

The power consumption estimates discussed in the main text, both for the memristor-based approach and for the GPU control, are specific to the inference phase. Training is performed off-chip on a GPU server, and subsequently, the network parameters are transferred to all the chips.

Corresponding changes to the manuscript: We have added “inference” wherever talking about energy numbers.

Comment 11:

Minor comments: There are typos and grammar errors. - The caption of Fig. 2 also needs to be revised. Figures 2 (a), (b), (d) and (e) are TEM images, not SEM images. - In Fig. 5c on page 5, ‘exprimental’ -> ‘experimental’, above the graph. - In line 172 on page 7, ‘plotted in black’ -> ‘plotted in red’ - In supplementary Fig.6, ‘To each distribution correspond 2 programming conditions ’ -> ‘Each distribution corresponds to 2 programming conditions ’, ‘The three currents give an activation’, and ‘The activation gives three input patterns ’

We have reviewed and corrected all these typos, we thank the reviewers for catching them!

Reviewer 2 (Remarks to the Author)

General comments

This manuscript reports the implementation of Bayesian neural network using memristor-based electronics, a critical improvement toward efficiently implementing Bayesian inference based on the authors' previous work (1. Dal ex-situ). As a demonstration, such electronics is used to process classification of heartbeats and estimate the certainty of predictions. This approach has advantage of lower energy consumption comparing to conventional CMOS electronic implementations.

We thank the reviewer for his/her review and these comments.

Comment 1:

This is not the first paper about using memristors to implement Bayesian neural network. Previous papers [2 Lin BNN real][3 Li Enabling] and the authors' previous work [4, Dalgaty harnessing res rand...] proposed to exploit different sources of variability in memristors to implement Bayesian inference. It is important to cite them in the state of the art and illustrate the novelty of this paper, if possible, compare with them.

We would like to thank the reviewer for bringing these works to our attention, which explore the utilization of cell-to-cell variability in populations of memory devices for storing Bayesian probabilistic weights. In response to this valuable feedback, we have entirely rewritten the introduction highlighting the novelty of our work more explicitly and incorporating a more extensive discussion the state of the art, including these article. Additionnally, we have included a comparison as a more detailed comparison between our work and the state of the art in a new Supplementary Note 14. We summarize its content here.

In the existing literature, several works have also investigated the potential of using memristors to store Bayesian probabilistic weights. We have compiled a comprehensive list of these works in Table 2, presenting a side-by-side comparison with our own work. Notably, our experimental Bayesian neural network stands out as the only system capable of performing on-chip inference, thanks to a novel hardware-calibrated training algorithm based on a variation of the popular Bayes By Backprop method. Conversely, other works in the state of the art primarily relied on computer simulations calibrated using experimental data. Furthermore, we detected out-of-distribution samples for the first time through the evaluation of epistemic uncertainty.

	This work	[1,4]	[2]	[3]
Technology	RRAM HfO_2 PCM GST	RRAM HfO_2	RRAM TaO_x/HfO_x	RRAM
Mechanism for probabilistic distribution construction	Population of devices	Population of devices	Population of devices + read to read	Population of devices + read to read
Training Algorithm	Hardware calibrated training (Bayes By Backprop + technological loss)	Markov Chain Monte Carlo	Bayes By Backprop	Stochastic Weight Averaging Gaussian
Scalability	Yes	No	Yes	Yes
Inference on Chip	Yes	No	No	No
Uncertainty Evaluation	Yes	No	Yes	Yes
Out of Distribution Detection	Yes	No	No	No

Table 2. Comparison of our work with approaches of the literature using memristors to store Bayesian probabilistic weights.

Comment 2:

The novelty of the research is not strong, in other words, not displayed. For example, the first innovation of this work, that is, training Bayesian neural network using variational inference, has already been reported for several times [2, 3]. And authors

should provide some comment on the computational efforts/cost of the off-chip training phase.

We have now entirely rewritten the introduction of paper to highlight the novelty of our work more explicitly. The key novelty – which allowed our experiments to work – was the introduction of a “technological loss” in variational inference to perform hardware-calibrated training, which deviates from the existing approach of using classical variational inference for off-chip training. During the training process, the technological loss term, $-\log(U_\Gamma(\theta))$, is incorporated into the overall loss function

$$Loss = Loss_{VI} - \log(U_\Gamma(\theta)). \quad (4)$$

The term $Loss_{VI}$ corresponds to the standard loss used in Bayes By Backprop, while $U_\Gamma(\theta)$ is determined by the experimental data, θ_{exp} , with the aim of approximating a uniform function over Γ while ensuring differentiability at the boundary.

$$U_\Gamma(\theta) = \tanh(\beta \cdot f(\theta)), \quad (5)$$

where $f(\theta)$ is defined as

$$f(\theta) = \frac{1}{\delta\sqrt{2\pi}} e^{-\frac{(\theta-\theta_{exp})^2}{2\delta^2}}, \quad (6)$$

where $\theta = (\mu, \sigma)$ and θ_{exp} is the nearest experimental point to θ achieved in hardware (i.e., presented in Fig. ?? of the main article). The parameters δ and β control the rate of increase of the technological loss outside Γ and the speed at which the technological loss approaches a minimum value close to the experimental points, respectively. When the value of $\theta = (\mu, \sigma)$ is significantly different from the closest experimental value θ_{exp} , the function $f(\theta)$ approaches zero, resulting in a large value for the technological loss. This has the effect of penalizing such values of θ and encourages the network to decrease the overall loss by bringing θ closer to θ_{exp} . Conversely, when θ is sufficiently close to θ_{exp} , $f(\theta)$ is large enough to cause U_Γ to saturate to 1, thereby resulting in a null technological loss. In such cases, the network is rewarded for such values of θ and can reduce the overall loss according to the standard rules of Bayes By Backprop. This balance between the use of Bayes By Backprop and the incorporation of the technological loss is achieved in a continuous and smooth manner, ensuring that the network can effectively learn and optimize while taking into account the technological constraints. The novel proposed hardware-calibrated training is fundamental to guarantee good performances in terms of both accuracy and uncertainty estimation (see Figure 8).

In order to compare the computational effort of the proposed hardware-calibrated training with the classical Bayes by Backprop method, we conducted a series of experiments to measure the time required for 100 epochs on a batch of 100 images using each approach. Our findings indicate that the network trained with the classical Bayes by Backprop method completed the training process in 22 seconds, while the hardware-calibrated training algorithm took 120 seconds. This implies that our model’s training process is approximately six times more computationally demanding than the classical Bayes by Backprop approach. However, it is crucial to emphasize that this training phase is a one-time requirement for the model. Once the model is trained, it can be deployed on any chip for inference at the edge.

We now included this information in the Methods section. Furthermore, considering this comment and similar ones, we have revised the introduction to better elucidate the novelty of our research and provide a more comprehensive comparison with the existing state of the art.

Comment 3:

In the manuscript, when implementing the network, authors sample M weight values for each synapse based on its probability distribution, and program them to the M memory array. I think it is using $2 \times M$ memristors to store sample value of a weight. However, the authors claimed that using 2 memristors to implement each probability synapse weight. Please clarify this.

Thanks to this comment, we have realized that the initial version of the manuscript was sometimes using inconsistent language, calling a synapse sometimes the $2 \times M$ devices, and sometimes the two devices of a unique sample. We have followed the recommendation of the reviewer and, throughout the manuscript and its Supplementary Notes, we now call “synapse” the ensemble of samples (i.e., $2 \times M$ devices), and pairs of device weight samples or weight values, depending on context.

Comment 4:

The γ scaling factor and the technological loss term both allow hardware to implement the weight using a memristor. What will the network performance be like with or without the technological loss term? And it would be better to add some explanations about the definition of the γ scaling factor (Eq.4).

Definition of the γ scaling factor: The γ scaling factor (Eq. 4 in the Methods of the main manuscript) simply multiplies a constant to the Bayesian weights obtained after the off-chip training. This conversion is performed to transform the Bayesian weights from arbitrary units to microsiemens. Its value is obtained using a rigorous methodology by minimizing the Kullback-Leibler divergence between the experimental and simulated normal distributions in our reference arrhythmia detection task.

To clarify this methodology, we have overhauled the ‘‘Correspondance between weight and conductance’’ section of Methods in the revised version of the manuscript.

Importance of the technology loss: We have now included a new Supplementary Note 9, which compares network performance with and without the use of the technological loss. We summarize its content here.

The device physics of both filamentary memristors and phase change memory imposes limitations on the range of Bayesian weights that can be stored (see Figure 3e in the main text). These limitations result in significant performance losses when attempting to transfer a model trained off-chip using the classical Bayes By Backprop method onto hardware chips. The issue arises because the training algorithm may require a weight that cannot be implemented in hardware. To address this challenge, in this work, we proposed a hardware-calibrated training method that considers the technological constraints associated with memristor or phase change memory technology. By introducing a technological loss term, this approach restricts the domain of achievable weights to those that can be experimentally realized with the given technology. In this note, we show the impact of using this technological loss.

Figures 8a and b, described with more details in the new Supplementary Note 9, compare the network accuracy and uncertainty estimation obtained by training a Bayesian neural network with and without using the technological loss. The results confirm that the technological loss allows for important improvement in the network performances in terms of both accuracy and uncertainty estimation. We can also observe that without the technological loss, the degradation is greater for phase change memories than for filamentary results. This result was predictable as the technological constraints are greater for phase change memories than filamentary memristors (see Fig. 3 of the main article).

Figure 8. **a** Network performances obtained on a memristor based Bayesian network trained with the popular Bayes By Backprop method and the novel hardware calibrated training algorithm based on the technology loss. **b** Network performances obtained on a PCM based Bayesian network trained with the popular Bayes By Backprop method and the novel hardware calibrated training algorithm based on the technology loss. The training process was repeated ten times, and each inference was repeated 50 times with $M = 10$ samples. The final result represents the average performance obtained from the 500 inferences.

Comment 5:

For germanium-antimony-tellurium phase-change memories, it would be better to study how the network performance is after one hour, one day, and two weeks. For hafnium-oxide-based filamentary memristors, iterative programming techniques suffer from conductance relaxation [5], which could also cause conductance drift like phase-change memory devices. Also, device-to-device variability could also induce weight deviation. Their impacts on network performance should also be investigated. It would be good if the authors could provide single-weight plots of target and measured distribution.

Thank you for these excellent recommendations. We have included two new Supplementary Notes (5 and 6), based on new measurements and simulations, to address them. We summarize their main content here.

Conductance relaxation in hafnium oxide-based memristors. Figure 9 compares the probability density of 2,048 memristors after relaxation (six seconds after iterative programming) with the target conductance values. Relaxation causes a rapid spread in the conductance of the devices, typically occurring within a few seconds after programming⁷. Figure 10 shows the technologically plausible domain of the normal distributions, $\Gamma_{memristor}$, over a span of two weeks. These results confirm that the conductance distributions remain stable after the initial six seconds. For this reason, we used the conductance distributions after relaxation as experimental data, denoted as $\theta_{exp} = (\mu_{exp}, \sigma_{exp})$, to define the technological loss term (Eq. 3 in Comment 3) to ensure a stable programming of the Bayesian neural networks.

Figure 9. Probability density of 2,048 memristors after relaxation (6 seconds after iterative programming) and the target conductance values used during the iterative programming.

Figure 10. Domain of the normal distributions ($\Gamma_{memristor}$) measured at different times after programming. **a** Right after relaxation at $t=6$ second and at $t=1$ hour, **b** at $t=1$ hour and at $t=1$ week, **c** at $t=1$ week and at $t=2$ weeks.

Network performances over time, for hafnium oxide-based memristors and phase change memories. To study the evolution of network performance over time, we conducted simulations using calibrated experimental data from Figure 10 for memristors and Figure 5 in Supplementary Note 4 for phase change memory. Since the effects of phase change memory drift and memristor conductance relaxation become negligible after one day and after six seconds, respectively, the technological loss term is based on the technologically plausible domain of normal distributions obtained after one day for phase change memory and after six seconds for memristors. The simulations involve two separate training processes, one for memristors and another for phase change memory. Inference simulations were repeated 50 times, and the error bars on the bar plots represent one standard deviation. Importantly, our results, reported in Fig. 11, demonstrate that the Bayesian Neural Network exhibits remarkable resilience to challenges posed by conductance relaxation and drift.

Figure 11. Network performances in terms of accuracy and uncertainty estimation over time for Bayesian neural network based on memristors **a** and PCMs **b**.

Comment 6:

The demonstrated network models (32x16x9) and the dataset (ECG data) are quite simple. In order to show the effectiveness of the proposed method more convincingly, more complex simulated or experimental network models and datasets (e.g., MNIST/CIFAR10) are required.

In the main body of the paper, our focus was to experimentally demonstrate our idea, which led us to present it using a selected simple dataset suitable for the size of the hardware. However, it is indeed crucial to demonstrate the scalability of our approach to larger datasets with more complex models. As suggested, we have included new results based on the MNIST and CIFAR datasets in the new Supplementary Notes 11 and 12, respectively. We summarize their main results here.

For this purpose, we first simulated, using the simulator validated in Supplementary Note 2, a two-layer *float32* deterministic convolutional neural network followed by a two-layer (1813, 128, 10) fully connected Bayesian neural network to address the MNIST dataset. The training of the Bayesian layers was calibrated on the experimental data from memristors using the technological loss described in the main article. The trained model achieved an accuracy on the MNIST test dataset of 99.2% with 50 samples using experimental distribution achieved with two filamentary memristors per sample.

To evaluate the epistemic uncertainty performances, we incorporated into our test dataset the Kuzushiji-MNIST (KMNIST) dataset¹⁰, as shown in 13b, which includes ten Hiragana characters in the MNIST format, and naturally provides unseen data. Figures 12c and d depict the probability density distributions of the aleatoric and epistemic uncertainty, respectively, using the same format as Fig. 5 of the main article. Different colors are used to represent correct predictions (blue), incorrect predictions (orange), and unseen data (red). We observed that the aleatoric uncertainty is lower than 0.25 for 99% of correctly classified data points, while it exceeds 0.25 for 81% of incorrectly classified data points and unseen data points. Additionally, 91% of the unseen images exhibit epistemic uncertainty higher than 0.25. These results demonstrate the capability of the Bayesian neural network to identify new and unknown images in the MNIST case.

Figure 12. Uncertainty estimation on MNIST dataset. **a** Images used for training. **b** Images used for out of distribution detection. **c** Probability density distribution of the aleatoric uncertainty for correct predictions, incorrect predictions and unseen diseases. **d** Probability density distribution of the epistemic uncertainty for correct predictions, incorrect predictions and unseen diseases.

Then, we evaluated our approach on the CIFAR-10 image recognition dataset¹¹, illustrated in Figure 13a. Our focus was specifically on evaluating our method for the last fully connected layers and used convolutional layers pretrained on ImageNet. We resized CIFAR images from 32x32 to 220x220 pixels. Additionally, we employed horizontal flipping to augment the dataset, effectively doubling its size. First, we applied the convolutional layers of a deterministic ResNet-18¹² network pretrained on ImageNet (available in the PyTorch library) to reconstruct a features-based dataset. Next, we used a two-layer (512, 512, 10) fully connected Bayesian neural network to classify CIFAR-10 features. The training was calibrated on the experimental data from memristors using the technological loss described in the main article. The trained model achieved an accuracy of 88% with 50 samples using experimental distribution achieved with two filamentary memristors per sample. We used the simulator validated in Supplementary Note 2,

To evaluate the epistemic uncertainty performances, we added unseen data to our test dataset. We used the data of the flower category of the CIFAR-100 dataset, shown in Figure 13b, which is not present in CIFAR-10. Figures 13c and d present the calculated aleatoric and epistemic uncertainties for correct predictions (blue), incorrect predictions (orange), and unseen data (red), plotted in the same format as Fig. 5 in the main body text. The aleatoric uncertainty is lower than 0.5 for 82% of all correctly classified data points, while it is higher than 0.5 for 75% of all incorrectly classified data points and unseen images. 89% of the unseen images have an epistemic uncertainty higher than 0.25. These results indicate that the Bayesian neural network can evaluate uncertainty and identify new unknown images in the complex CIFAR dataset.

Figure 13. Uncertainty estimation on CIFAR dataset. **a** Images used for training. **b** Images used for out of distribution detection. **c** Probability density distribution of the aleatoric uncertainty for correct predictions, incorrect predictions and unseen diseases. **d** Probability density distribution of the epistemic uncertainty for correct predictions, incorrect predictions and unseen diseases.

Comment 7:

The manuscript lacks benchmark and comparison with other in-memory circuit implementation, such as [1][2][3][4], which could significantly improve the impact of this work.

We have now incorporated a new Supplementary note 14 benchmarking our approach with these works. We have also included the main elements of this comparison in our rewritten version of the Introduction. We described these changes in our answer to Comment 1 of Reviewer 2.

Comment 8:

It would be better if the authors could improve the organization of this paper, especially introduction section to highlight the novelty. It would be good if the authors could strengthen that energy consumption estimation section.

Based on this comment and other comments from reviewers one and two, we have completely overhauled the introduction section to highlight the novelty more prominently. We have also considerably improved the introduction to Bayesian neural networks and to our memory-based implementation. Based on this comment and other comments of reviewers 2 and 3, we have entirely redone the whole energy consumption estimation, based on better and more rigorous analysis. Its main findings are described in the main body text, and its details in the new Supplementary Note 15. We hope the reviewers will like our revised manuscript.

Comment 9:

Minor comments In 1, based on line 172 while the unseen disease data points are plotted in black might be while the unseen disease data points are plotted in red.

We have corrected this error, thank you for catching it.

Reviewer 3 (Remarks to the Author)

General Comments:

The authors proposed a hardware-based Bayesian neural network (BNN) by employing the intrinsic random resistance distribution of memristors (filamentary oxide memristor or phase change memory) for edge applications, such as medical diagnoses. Some specific designs are reported: (1) variational inference; (2) the synapse is evaluated using two memristors; (3) A critical “technological loss” is introduced. In the application of classifying heartbeats signals, the proposed BNN shows comparable accuracy, aleatoric and epistemic uncertainty estimations to software simulations. My main concern about this manuscript is the novelty. The same team has already reported several studies combining Bayesian theory with memristors. For example, in the work (“A memristor-based Bayesian machine, *Nature Electronics*, 6, 52–63, 2023”), the authors used a memristor array to implement the Bayesian inference where the probability is encoded as a bit stream. In this manuscript, the authors describe the difference between this work and the prior work as “Ref.18 exploited the probabilistic nature of memristors to perform Bayesian learning. This approach can only be applied to small-scale tasks, but unlike Bayesian neural networks, it does not suffer from the limitations imposed by the correlation of mean value and standard deviation of memristors”. However, I notice that there are substantial similarities as follow: (1) Both are dedicated to energy-efficient edge applications (2) Both works focus on using the random resistance of memristor to implement the Bayesian inference/reasoning (3) Both have similar architecture (4) Same programming method: compare or add/subtract two memristor units to get random values or conductance. In my opinion, this manuscript is an incremental improvement of the previous work, and the improvement is insufficient for publication in the *Nature Communications*.

We thank the reviewer for his/her review and these comments.

Comment 1:

A major issue of this work is related to the number of neural networks and synaptic weight, which causes a high energy consumption: 270 nJ/inference. It is better than the conventional GPU hardware, but not competitive with memristive neural networks. For edge applications, power consumption is a key concern. Can the authors demonstrate how to overcome this shortcoming, and balance the necessity of uncertainty prediction and the power consumption? Apart from BNNs, a lot of work has been done to predict uncertainty in ANNs. Besides BNN, is it possible to use memristors to implement a high energy-efficient ANN with uncertainty prediction?

We agree with the reviewer that implementing Bayesian neural networks is more challenging compared to ANN: Bayesian neural networks involve repeated sampling and feed-forward computing. However, ANNs lack uncertainty quantification. They often exhibit incorrect yet overconfident predictions, which can lead to catastrophic consequences in safety-critical systems like robots, autonomous vehicles, or clinical practice¹³.

Still, two principal methods have been developed for estimating uncertainty in non-Bayesian artificial neural networks (ANNs), and it is very interesting to think if they could be used for a memristor-based hardware implementation. The first, deep ensembles, trains multiple identical ANNs, creating a prediction distribution but offering no energy or hardware benefits¹⁴. Moreover, implementation challenges arise when transferring high-precision parameters into the imprecise conductance states of resistive memory in memristor-equipped ANNs. In contrast, Bayesian neural networks adeptly exploit memristor variability to store random variables, making them ideal for resistive memory-based hardware.

The second method, Monte Carlo (MC) dropout, generates a prediction distribution by randomly disabling nodes within the model^{15,16}. However, the requirement for multiple forward passes precludes any reduction in energy consumption, and it is not natural to implement in a memristor-based circuit.

Thus, memristor-based Bayesian neural networks emerge as the more promising approach for both efficient uncertainty quantification and energy conservation via analog in-memory computation.

We have included these new elements in the Discussion section of the revised version of the manuscript.

Concerning energy consumption, we have entirely overhauled this point of our manuscript. All these changes are described in our response to Comment 4 of Reviewer 3.

Comment 2:

The domains of normal distribution of filamentary memristor and phase-change memory are quite different. How does this distribution affect the accuracy, aleatoric and epistemic uncertainty estimations? It is stated that “To extend the domain of normal distributions, we store each sample of a probabilistic weight as the difference between the conductance values of two adjacent memory cells”. Can we further extend the domain of normal distributions by using more memory cells? What are the pros and cons? If one memory cell is used, do the accuracy, aleatoric and epistemic uncertainty estimations deteriorate?

We appreciate these suggestions, which emphasizes the significance of the size of the domain of normal distributions on network performance in terms of both accuracy and uncertainty estimation.

Impact of the distribution on the performance in terms of accuracy and uncertainty evaluation: The domain of normal distribution obtained using filamentary memristors surpasses the one achieved in terms of both σ and μ (refer to Figure 3e in the main text), thereby enabling improvements in network performance in both accuracy and uncertainty estimation (see the updated Table 1 in the main text). In the revised version of the manuscript, we discuss this point more explicitly.

What happens if we use more than two devices, or only one device: To answer these questions, we have obtained new results, presented in the new Supplementary Note 7. We summarize its main findings here.

Figure 14a illustrates the domain of normal distributions obtained using one (blue), two (green), and four (yellow) devices per sample of a Bayesian probabilistic weight. Increasing the number of devices per sample expands the attainable size of the domain of normal distributions in experimental settings. As depicted in Figure 14b, a wide range of possibilities for σ and μ enhances both neural network accuracy and uncertainty estimation. A broad spectrum of possibilities for μ is crucial for achieving high accuracy, while a diverse range of possibilities for σ is necessary for accurate uncertainty estimation. It is important to keep σ small to ensure minimal output variability for well-known inputs. Conversely, σ needs to be large to provide output variability for unknown input data, thereby facilitating uncertainty estimation.

Figure 14. Impact of the number of memristors per sample of a Bayesian probabilistic weight **a** Domain of the normal distribution obtained experimentally by storing samples on one (blue), two (green), and four (yellow) memristors. μ and σ are normalized to the minimum achievable standard deviation, σ_{min} , obtained with different experimental conditions. **b** Network performances obtained by storing samples on one, two, and four memristors. The training process was repeated ten times, and each inference was repeated 50 times with $M = 10$ samples. The final result represents the average performance obtained from the 500 inferences.

Comment 3:

BNNs provide a high prediction confidence, but the classification accuracy is around 75 percent, which is much lower than other reported memristor-based neural network for the similar bio-signal classification (95percent, *Sci. Adv.* 2020; 6: eabc4797; *Nat Commun* 11, 4234, 2020). What is the reason for the poor accuracy? Can it be improved to a comparable level?

The poor accuracy reported in our work can be attributed to the inherent challenges associated with the chosen task, which is more complex than the bio-signal classification proposed in^{17,18}, as it has a lot of ambiguity. We made this choice to emphasize the uncertainty-evaluation capability of our system.

In our system, we have developed a classification model capable of differentiating among nine types of heart arrhythmia, which is more challenging than the three classes distinguished by the system proposed in¹⁸. To further evaluate the capabilities of our system, we intentionally concealed one class during the training process. As a result, approximately 7% of the test dataset became unclassifiable. This deliberate setup allowed us to assess the network's ability to recognize unfamiliar patterns and quantify the associated epistemic uncertainty. The difficulty of the chosen task is underscored by the performance of

a conventional neural network with the same architecture and using *float32* encoding for the synapses, which achieved a classification accuracy of only 80% on the same task.

To further validate the effectiveness of the proposed neural network, we simulated the implementation of a two-layer (64, 32, 3) fully connected Bayesian neural network using our simulator (Supplementary Note 2), designed to address the same task as the one presented in¹⁸. To extract the 64 input features required for our model, we applied a Fast Fourier Transform. Our trained model achieved a high accuracy of 94%, which closely aligns with the results reported in the referenced paper.

To clarify this point in the revised manuscript, we added a comment in the Discussion section of the manuscript about the reason for a relatively low accuracy, and incorporated the simulated performance of our approach on the dataset of¹⁸.

Comment 4:

The performance of the memristive neural network should be added to Table 1 to make the comparison more convincing. The authors claim 800 times improvement in energy efficiency comparing to the conventional GPU platform. However, the benchmarks and the comparison method for GPU and memristor BNN are unclear. More details should be provided. It will make more sense to compare at a system level rather than focusing on the processing part.

We have now entirely overhauled the energy analysis of our paper. Upon revisiting our analysis, we indeed identified inaccuracies in our preliminary projections of the energy consumption for the memristor-based circuit. These estimations were initially derived from published results, which we had not properly understood, and which led us to considerably overestimate the energy consumption number. We deeply regret the oversight and have since undertaken a comprehensive revision of these projections, using several, instead of just one, sources, ensuring the accuracy and reliability of our data.

Simultaneously, we re-evaluated our control mechanism, which initially utilized a GPU. We realized that this benchmark was not as useful as we hoped. Due to the high degree of parallelism of GPU, their energy consumption scales in non-obvious way with the size of a Bayesian neural networks and the batch size. Also, a large GPU is not a device that would be used for the extreme-edge applications that we are targeting, but rather microcontroller units (MCUs). Typically, STMicroelectronics STM32 MCUs are used for edge AI. MCUs are simpler devices than GPUs, and it is much easier to interpret measurements of their energy consumption meaningfully. Therefore, we now implemented the computation of our reference Bayesian neural networks on an STM32 and use this as our main control. While these modifications have led to substantial changes in the technical aspects of our manuscript, the overall conclusion remains largely consistent. We still posit that memristor-based Bayesian neural networks offer an efficient approach to uncertainty quantification and energy conservation.

The new energy analysis is spread between the Discussion and the Methods sections of the revised manuscript and a new Supplementary Note 15. We summarize its main findings here.

To estimate the energy consumption of the Bayesian neural network we first calculated the number of dot product operations for one inference:

$$Operations = 4 \cdot I_l \cdot H_l + 4 \cdot H_l \cdot O_l. \quad (7)$$

Here I_l is the input length, H_l is the hidden layer length, and O_l is the output length. The factor four is due to fact that each sample of a Bayesian probabilistic weight is stored as the difference between the conductance values stored in two memory cells and that a dot product contains addition and multiplication. One inference costs 2.624 operations. The cost of a single analog Multiply-and-Accumulate (MAC) operation in a resistive memory-based analog in-memory computing circuit depends on the input and output size and on the weight precision, and can vary considerably depending on the memory technology, CMOS node, array size, and design choices. We relied on energy per operation number of three industrial platforms employing resistive memory^{19,20}, and phase change memory²¹. The results are reported in Supplementary note 15 and Table 3. We found a cost ranging between 0.7 and 2.5 nanojoules per inference. Note that these estimates consider only the Multiply-and-Accumulate, which we expect to dominate. Still, additional circuitry will be needed, e.g., to present the input, analyze the outputs, and transfer data between arrays of the neural network.

To gain a perspective on the energy efficiency of the proposed approach compared to conventional hardware, we benchmarked this figure to the energy required for running the operations to perform inference of the same Bayesian neural network on an STM32F746ZGT6 MCU (integrated on a test Nucleo-F746ZG board), which is typically used for edge AI applications. These operations coded in the C language using the ST Microelectronics STM32 Cube integrated development environment and compiled and built without debugging options and using the strongest optimizations for speed (-Ofast option). To provide a fair comparison with our in-memory-computing platform, our C code includes only the multiply-and-accumulate operation. (We controlled that multiply-and-accumulate operations represented more than 99% of the execution time of our program.) We timed our program, and measured the current consumption of the MCU using an Ampere meter (we measured the current solely consumed by the MCU, excluding any other component of the board). The STM32F746ZGT6 MCU is fabricated in

a 90-nanometer CMOS node. We found a consumption of 170 microjoules per inference (with ten samples of the Bayesian neural network).

	Wan et al. Nature 2022 ²⁰	Cheng-Xin Xue et al. Nat. Electron. 2021 ¹⁹	Khaddam et al. IEEE JSSC 2022 ²¹
	HfOx/TaOx		GST
Device	RRAM	RRAM	PCM
CMOS node	130nm	22nm	14nm
Input bit width	4b	4b	8b
Output bit width	6b	11b	8b
Reported energy efficiency (TOPS/W)	16	36.61	10.5
Estimation of the energy in inference for the Bayesian hardware with N=10	1640 pJ	720 pJ	2500 pJ

Table 3. Comparison of the energy efficiency estimated in several state-of-the-art in-memory computing circuits based on memristors and PCMs

References

1. Blundell, C., Cornebise, J., Kavukcuoglu, K. & Wierstra, D. Weight uncertainty in neural network. In *International conference on machine learning*, 1613–1622 (PMLR, 2015).
2. Fortunato, M., Blundell, C. & Vinyals, O. Bayesian recurrent neural networks. *arXiv preprint arXiv:1704.02798* (2017).
3. Dalgaty, T. *et al.* In situ learning using intrinsic memristor variability via markov chain monte carlo sampling. *Nat. Electron.* **4**, 151–161 (2021).
4. Joshi, V. *et al.* Accurate deep neural network inference using computational phase-change memory. *Nat. communications* **11**, 1–13 (2020).
5. Tsai, H. *et al.* Inference of long-short term memory networks at software-equivalent accuracy using 2.5 m analog phase change memory devices. In *2019 Symposium on VLSI Technology*, T82–T83 (IEEE, 2019).
6. Dalgaty, T., Esmanhotto, E., Castellani, N., Querlioz, D. & Vianello, E. Ex situ transfer of bayesian neural networks to resistive memory-based inference hardware. *Adv. Intell. Syst.* **3**, 2000103 (2021).
7. Esmanhotto, E. *et al.* High-density 3d monolithically integrated multiple 1t1r multi-level-cell for neural networks. In *2020 IEEE International Electron Devices Meeting (IEDM)*, 36–5 (IEEE, 2020).
8. Sebastian, A. *et al.* Two-dimensional materials-based probabilistic synapses and reconfigurable neurons for measuring inference uncertainty using bayesian neural networks. *Nat. communications* **13**, 1–10 (2022).
9. Liu, S. *et al.* Bayesian neural networks using magnetic tunnel junction-based probabilistic in-memory computing. *Front. Nanotechnol.* **78** (2022).
10. Clanuwat, T. *et al.* Deep learning for classical japanese literature (2018). [cs.CV/1812.01718](https://arxiv.org/abs/1812.01718).
11. Krizhevsky, A. & Hinton, G. Learning multiple layers of features from tiny images. Tech. Rep. 0, University of Toronto, Toronto, Ontario (2009).
12. He, K., Zhang, X., Ren, S. & Sun, J. Deep residual learning for image recognition. In *Proceedings of the IEEE conference on computer vision and pattern recognition*, 770–778 (2016).
13. Dolezal, J. M. *et al.* Uncertainty-informed deep learning models enable high-confidence predictions for digital histopathology. *Nat. communications* **13**, 6572 (2022).

14. Lakshminarayanan, B., Pritzel, A. & Blundell, C. Simple and scalable predictive uncertainty estimation using deep ensembles. In *Advances in Neural Information Processing Systems 30 (NIPS 2017)* (2017).
15. Gal, Y. & Ghahramani, Z. Dropout as a bayesian approximation: Representing model uncertainty in deep learning. In *Proceedings of The 33rd International Conference on Machine Learning, PMLR 48:1050-1059, 2016* (2016).
16. Sida Wang, a. M. Fast dropout training. In *Proceedings of the 30th International Conference on Machine Learning, PMLR 28(2):118-126, 2013* (2013).
17. Liu, Z. *et al.* Multichannel parallel processing of neural signals in memristor arrays. *Sci. advances* **6**, eabc4797 (2020).
18. Liu, Z. *et al.* Neural signal analysis with memristor arrays towards high-efficiency brain–machine interfaces. *Nat. communications* **11**, 4234 (2020).
19. Xue, C.-X. *et al.* A cmos-integrated compute-in-memory macro based on resistive random-access memory for ai edge devices. *Nat. Electron.* **4**, 81–90 (2021).
20. Wan, W. *et al.* A compute-in-memory chip based on resistive random-access memory. *Nature* **608**, 504–512 (2022).
21. Khaddam-Aljameh, R. *et al.* Hermes-core—a 1.59-tops/mm² pcm on 14-nm cmos in-memory compute core using 300-ps/lsb linearized cco-based adcs. *IEEE J. Solid-State Circuits* **57**, 1027–1038 (2022).

Review on the revised manuscript “**Bringing uncertainty quantification to the extreme-edge with memristor-based Bayesian neural networks**” authored by Djohan Bonnet *et al.*, submitted to *Nature Communications*. [Manuscript Number: NCOMMS-23-00214A]

The manuscript has been effectively reorganized, highlighting the main novelty of this work. In the introduction section, a well-explained description was provided regarding the limitations of conventional neural networks and the advantages and operational principles of Bayesian neural networks. In addition, various additional measurement results and theoretical explanations were included to support the performance of this study. Moreover, detailed explanations regarding the essential novelty of this research, namely “technological loss,” were provided. However, this reviewer suggests a few additional modifications to improve the manuscript further.

1. Figure 1b was revised to show the number of output neurons and output currents accurately. Still, it would be better to indicate the number of input and hidden neurons using omission marks.
2. In line 197 on page 8, while explaining Figure 5d, it was mentioned that since 98 % of the unseen disease data points have epistemic uncertainty greater than 0.5, new unknown inputs can be well distinguished. However, in Supplementary Note 12, it was mentioned that since 89 % of the unseen images have epistemic uncertainty greater than 0.25, new unknown images can be identified in the complex CIFAR dataset. It is necessary to clarify the criteria for determining the threshold values of epistemic or aleatoric uncertainty that enable the distinction of unknown data in each application.
3. In Supplementary Figure 3, the descriptions corresponding to Fig.3a, 3b, and 3c should be placed directly following the labels (a), (b), and (c).
4. Related comment 3: The presented data only displays cycle-to-cycle and cell-to-cell variation for a single programming level in memristors and PCM. It would be advantageous to present cycle-to-cycle and cell-to-cell variation for all eight programming levels in memristors and phase change memories to show that the Bayesian neural network works well even with these variations.

5. In Supplementary Figure 6a, it is necessary to incorporate an additional data point at $t = 0$ (sec) to illustrate the initial conductance relaxation in memristors. Moreover, the conductance distributions of memristors after the initial six seconds, as depicted in Supplementary Figure 6a and b, do not exhibit stabilization, in contrast to the findings presented in Supplementary Figure 5c. Consequently, it is essential to comprehensively explain of conductance distribution observed at $t > 6$ (sec).
6. Minor comments: There are some typos and grammar errors.
 - In Fig. 5b on page 10, 'inorrect' → 'incorrect' on the graph.
 - In line 322, '**Algorithm 2**' on page 12, 'Voltage incremment' → 'Voltage increment'
 - In supplementary Fig. 6a '6 Second' → '6 Seconds'
 - In supplementary Fig. 6b and 6c, '1 Weeks' → '1 Week'
 - In supplementary Fig. 8b, '2 pair' → '2 pairs'
 - In supplementary note 11, on the second line of the third paragraph on page 11, the description of Kuzushiji-MNIST (KMNIST) states 'as shown in Supplementary Figure 13b'. However, the correct figure for KMNIST is 'Supplementary Figure 12b'. Please update the reference accordingly.

Reviewer #2 (Remarks to the Author):

I appreciate the clarifying comments from the authors and the substantial work in the additional analyses. I'm pleased to see that my previous concerns have been addressed in this revision. And the authors have extensively revised the manuscript and improved the overall quality. I would like to recommend that the paper be accepted for publication.

Reviewer #3 (Remarks to the Author):

The authors have addressed many of my technical comments and improved the overall clarity and quality of the manuscript. However, I still have a few concerns that need to be addressed.

(1) In my previous review, I raised a concern about the novelty of the research, which the authors did not fully address. Recently, in the paper titled "A memristor-based Bayesian machine" published in Nature Electronics (6, 52–63, 2023), the authors investigated a memristive Bayesian machine using a similar memristor-based 1T1R array, with a similar weight presentation method (using two memristor units to represent 1 value). It seems that both works are based on the same circuit and architecture, but applied to different applications. The authors should explicitly explain the differences between the two works to clarify the foundational differences in circuit design and programming schemes. Without addressing these differences, this work might be perceived as an incremental improvement of the previous research, which could raise concerns about its suitability for publication in Nature Communications.

(2) Regarding the energy analysis, I understand the authors' point about the inappropriate comparison between the memristive in-memory computing circuit designed for edge computing and a large GPU. However, comparing it with STM32F4, which is a typical MCU designed for embedded system control and not specifically intended for edge AI purposes, might not be fair either. The STM32F4 operates as a CPU performing computing in a serial manner, whereas memristive circuits perform computing in parallel, requiring significantly less time than a CPU. This difference in computing schemes makes it difficult to observe the advantages of memristors for BNN implementation. I recommend using edge GPUs (such as Jetson Nano) or MCUs with accelerators (such as Gap9 from Greenwaves) as new baselines to provide a fairer comparison.

(3) In this version, the authors removed the power estimation and instead provided typical power consumption from previous works, indirectly illustrating the power advantages of memristors and PCM chips. While this information is useful, it would be beneficial to include a more detailed analysis of the power efficiency in the revised manuscript.

(4) As mentioned in the rebuttal letter, I agree that the uncertainty evaluation task is more challenging than classification tasks. The slow modeling of uncertainty is a main drawback of BNNs, and if the authors' method can accelerate BNNs, it could be a potential solution. However, it is

essential to compare the proposed solution with well-established deep learning algorithms and discuss their respective advantages and limitations. I suggest referencing a few works on this topic, such as "Bounding Box Regression with Uncertainty for Accurate Object Detection" (CVPR 2019) and "Gaussian YOLOv3: An Accurate and Fast Object Detector Using Localization Uncertainty for Autonomous Driving" (ICCV 2019), to provide a more comprehensive comparison.

Response to Reviews

We would like to thank the anonymous reviewers for their time and excellent feedback, which has allowed us to improve the quality of our manuscript. We have addressed the points raised by the reviewers and revised the manuscript accordingly. The revised versions includes additional experiments to investigate cycle-to-cycle and cell-to-cell variations across eight programming levels for both memristors and PCM technologies. We have also decided to include a benchmark using an NVIDIA Jetson Nano, a device capable of parallel computing.

Reviewer 1 (Remarks to the Author)

General comments

The manuscript has been effectively reorganized, highlighting the main novelty of this work. In the introduction section, a well-explained description was provided regarding the limitations of conventional neural networks and the advantages and operational principles of Bayesian neural networks. In addition, various additional measurement results and theoretical explanations were included to support the performance of this study. Moreover, detailed explanations regarding the essential novelty of this research, namely “technological loss,” were provided. However, this reviewer suggests a few additional modifications to improve the manuscript further.

We thank the reviewer for his/her review and these encouraging comments.

Comment 1:

Figure 1b was revised to show the number of output neurons and output currents accurately. Still, it would be better to indicate the number of input and hidden neurons using omission marks.

Based on this comment, we have modified the figure.

Comment 2:

In line 197 on page 8, while explaining Figure 5d, it was mentioned that since 98 % of the unseen disease data points have epistemic uncertainty greater than 0.5, new unknown inputs can be well distinguished. However, in Supplementary Note 12, it was mentioned that since 89 % of the unseen images have epistemic uncertainty greater than 0.25, new unknown images can be identified in the complex CIFAR dataset. It is necessary to clarify the criteria for determining the threshold values of epistemic or aleatoric uncertainty that enable the distinction of unknown data in each application.

The choice of the threshold value depends on the specific application and context. For instance, applications that necessitate zero false negatives will require a lower threshold. Due to this consideration, we opted not to utilize this metric for comparing our results with the existing literature. The Area Under the Curve (AUC) of the Receiver Operating Characteristic (ROC) is application-independent and thus more suitable for making comparisons. This is the metric we have adopted for Table 1.

We have clarified this point in the main text.

Comment 3:

In Supplementary Figure 3, the descriptions corresponding to Fig.3a, 3b, and 3c should be placed directly following the labels (a), (b), and (c).

Based on this Comment and Comment 4, we have made modifications to Figure 3 in the Supplementary Information, along with its corresponding caption.

Comment 4:

Related comment 3: The presented data only displays cycle-to-cycle and cell-to cell variation for a single programming level in memristors and PCM. It would be advantageous to present cycle-to-cycle and cell-to-cell variation for all eight programming levels in memristors and phase change memories to show that the Bayesian neural network works well even with these variations.

In response to this comment, we conducted additional experiments to investigate cycle-to-cycle and cell-to-cell variations across eight programming levels for both memristors and PCM technologies. Figure 1 illustrates the results, with distinct clusters of data points in Figures 1a and 1b representing different programming conditions. Each data point reflects the mean and standard deviation derived from programming 1,000 distinct devices, with varying points indicating 1,000 cycles per programming condition. These findings affirm the stability of device-to-device variability across multiple programming iterations under all studied conditions, whether with memristors or phase change memories.

Figures 1c and 1d compare the probability density of 1,000 distinct devices programmed once (representing device-to-device variability) with the probability density of one device cycled 1,000 times (representing cycle-to-cycle variability) for memristors and PCM technologies, respectively. For each technology, we have presented the distributions obtained using the programming conditions corresponding to the smallest and largest mean conductance values, as well as an intermediate distribution. Both sources of variability demonstrate a similar impact on conductance variability across different programming conditions.

Supplementary Note 3 and Supplementary Figure 6 have been updated to incorporate the new experiments examining cycle-to-cycle and cell-to-cell variations under various programming conditions.

Figure 1. Impact of cycle-to-cycle variability on cell-to-cell variations for memristors and PCMs. **a** Scatter plot illustrating the influence of cycle-to-cycle variability on cell-to-cell variation in memristors for different programming conditions. Different clusters of data points represent distinct programming conditions. Each data point reflects the mean and standard deviation of a Gaussian distribution obtained by programming 1,000 distinct devices, with varying points representing 1,000 cycles per each programming condition. **b** Scatter plot illustrating the effect of cycle-to-cycle variability on cell-to-cell variation in phase change memory for different programming conditions. **c** Probability density of 1,000 cells programmed under the same conditions, depicting device-to-device variability (blue), compared with the probability density of one device programmed 1,000 times (green) for memristor technology. The experiment has been repeated for three different programming conditions corresponding to the smallest and largest mean conductance values in **a**, as well as an intermediate distribution. **d** Probability density of 1,000 cells programmed under the same conditions, illustrating device-to-device variability (blue), compared with the probability density of one device programmed 1,000 times (green) for PCM technology. The experiment has been repeated for three different programming conditions corresponding to the smallest and largest mean conductance values in **b**, as well as an intermediate distribution.

Comment 5:

In Supplementary Figure 6a, it is necessary to incorporate an additional data point at $t = 0$ (sec) to illustrate the initial conductance relaxation in memristors. Moreover, the conductance distributions of memristors after the initial six seconds, as

depicted in Supplementary Figure 6a and b, do not exhibit stabilization, in contrast to the findings presented in Supplementary Figure 5c. Consequently, it is essential to comprehensively explain of conductance distribution observed at $t > 6$ (sec).

In reponse to this feedback, we incorporated an additional data point at $t = 0$ sec and compared it with data point at $t = 6$ sec (Figure 4). Rapid conductance spread, or short-term conductance relaxation, is observed immediately after programming, between $t = 0$ sec and $t = 6$ sec, accordingly with data presented in Figure 5 in Supplementary Information. This fast relaxation effect occurs on the same time scale throughout the conductance range. Conductance spread then continues at a slower pace over longer time scales as demonstrated in Figure 3, which plots the standard deviation of normal distributions obtained under three distinct programming conditions measured at varying times post-programming. This result aligns with our previous work¹. Consequently, the technological loss term is based on the technologically plausible domain of normal distributions obtained after six seconds for memristors. As demonstrated in Supplementary note 7, the long-term relaxation only slightly affects the performances of the neural network.

Supplementary Note 5 and Supplementary Figure 6 have been revised to include an additional data point at $t = 0$ sec and provide a more detailed explanation of both short-term and long-term relaxation phenomena.

Figure 2. Domain of the normal distributions ($\Gamma_{memristor}$) measured at different times after programming. **a** at $t=0$ second and right after relaxation at $t=6$ seconds, **b** Right after relaxation at $t=6$ seconds and at $t=1$ hour, **c** at $t=1$ hour and at $t=1$ week, **d** at $t=1$ week and at $t=2$ weeks.

Figure 3. Standard deviation of normal distributions obtained under three different programming conditions measured at different times post-programming.

Comment 6:

Minor comments: There are some typos and grammar errors. - In Fig. 5b on page 10, ‘inorrect’ -> ‘incorrect’ on the graph. - In line 322, ‘Algorithm 2’ on page 12, ‘Voltage incremment’ -> ‘Voltage increment’ - In supplementary Fig. 6a ‘6 Second’ -> ‘6 Seconds’ - In supplementary Fig. 6b and 6c, ‘1 Weeks’ -> ‘1 Week’ - In supplementary Fig. 8b, ‘2 pair’ -> ‘2 pairs’ - In supplementary note 11, on the second line of the third paragraph on page 11, the description of Kuzushiji-MNIST (KMNIST) states ‘as shown in Supplementary Figure 13b’. However, the correct figure for KMNIST is ‘Supplementary Figure 12b’. Please update the reference accordingly.

We thank the reviewer for this comment. We modified the Figures and the text accordingly.

Reviewer 2 (Remarks to the Author)

General comments

I appreciate the clarifying comments from the authors and the substantial work in the additional analyses. I'm pleased to see that my previous concerns have been addressed in this revision. And the authors have extensively revised the manuscript and improved the overall quality. I would like to recommend that the paper be accepted for publication.

We would like to express our heartfelt thanks to the reviewer for his/her favorable comments.

Reviewer 3 (Remarks to the Author)

General Comments:

The authors have addressed many of my technical comments and improved the overall clarity and quality of the manuscript. However, I still have a few concerns that need to be addressed.

We thank the reviewer for his/her review and these comments.

Comment 1:

In my previous review, I raised a concern about the novelty of the research, which the authors did not fully address. Recently, in the paper titled "A memristor-based Bayesian machine" published in Nature Electronics (6, 52–63, 2023), the authors investigated a memristive Bayesian machine using a similar memristor-based 1T1R array, with a similar weight presentation method (using two memristor units to represent 1 value). It seems that both works are based on the same circuit and architecture, but applied to different applications. The authors should explicitly explain the differences between the two works to clarify the foundational differences in circuit design and programming schemes. Without addressing these differences, this work might be perceived as an incremental improvement of the previous research, which could raise concerns about its suitability for publication in Nature Communications.

Thank you for the opportunity to clarify the differences between our work and the research presented in Ref.². While at a glance, both systems employ the same 2T2R memristor-based technology, there are foundational differences, both conceptually and practically, as outlined below:

- The Bayesian machine of Ref.² uses memristors as memory for the model parameters and uses stochastic computing to perform inference on a Bayesian network. Bayesian networks differ from Bayesian neural networks investigated in the present submission. The former are constructed using expert knowledge and are fully explainable, which makes them ideal for tasks like sensor fusion. On the contrary, Bayesian neural networks are trained from the ground up and excel on more data-intensive tasks like electrocardiogram or electroencephalogram classification.
- From a circuit point of view, the Bayesian machine also differs strongly from the present work: The Bayesian machine is a digital system that tolerates memristor imperfections but does not exploit them².

We have now rewritten the second-to-last paragraph of the introduction to position our work with regard to the Bayesian machine very explicitly. Also, we have included a detailed comparison between the present work and the Bayesian machine within Supplementary Note 14.

Comment 2:

Regarding the energy analysis, I understand the authors' point about the inappropriate comparison between the memristive in-memory computing circuit designed for edge computing and a large GPU. However, comparing it with STM32F4, which is a typical MCU designed for embedded system control and not specifically intended for edge AI purposes, might not be fair either. The STM32F4 operates as a CPU performing computing in a serial manner, whereas memristive circuits perform computing in parallel, requiring significantly less time than a CPU. This difference in computing schemes makes it difficult to observe the advantages of memristors for BNN implementation. I recommend using edge GPUs (such as Jetson Nano) or MCUs with accelerators (such as Gap9 from Greenwaves) as new baselines to provide a fairer comparison.

Thank you for this valuable input. Following this suggestion, we have included another benchmark using an NVIDIA Jetson Nano, a device capable of parallel computing. We evaluated energy consumption for various scenarios, as depicted in Figure 4. We performed inference using 1, 10, or 100 samples of the output per input, and also different batches (i.e., the number of inputs processed in parallel by the Jetson Nano GPU). During these measurements, we monitored the power consumption of both the NVIDIA Jetson Nano GPU and the whole system, using the built-in power monitoring feature of the board. Figure 4 presents the energy required to infer an input, in all these situations. Notably, larger batch sizes result in lower average energy consumption per input, due to the parallelization of computing. Additionally, multiplying the number of samples for a single inference by ten does not proportionally increase the energy required, again due to parallelization.

For ten samples, the configuration we previously used in our benchmark, we achieved energy consumption of less than ten microjoules per inference for batch sizes larger than 100. This energy consumption is an order of magnitude lower than the STM32 MCU, but still several orders of magnitude above the performance achievable with memristive devices. This result underscores the potential efficiency of memristive-based ASICs for edge computing applications.

Methodology. We have included another benchmark using an NVIDIA Jetson Nano, an edge-computing board widely used for edge AI applications equipped with an NVIDIA Tegra X1 system on chip, featuring a GPU and a multicore CPU. This chip-based system is manufactured in a more modern 20-nanometer CMOS node. In our benchmark test, we perform the multiply-and-accumulate operations of our system on the GPU. Our benchmark code is written using Pytorch 1.10 with NVIDIA Jetpack 4.6 and NVIDIA CUDA 10.2. All the multiply-and-accumulate operations for the different output samples are performed with a single tensor multiplication using the Pytorch torch.matmul function, ensuring a fully parallel operation and an optimal use of the GPU. Additionally, our code allows batching, i.e., the processing of several inputs simultaneously within the same torch.matmul call. As described in the Discussion section of the paper, higher batching allows a better use of the resources of the GPU and reduces the energy consumption per input. To obtain a reliable estimate of energy consumption, with repeated the multiply-and-accumulate operations multiple times and timed the process using the repeat function from the Python 1.10 timeit library. We chose the number repetitions to reach a total computation time of a minute, allows the power consumption of the board to stabilize. During these measurements, we monitored the power consumption of both the NVIDIA Jetson Nano GPU and the whole system, using the built-in power monitoring feature of the board. The energy consumption is obtained by multiplying the computation time of a torch.matmul call by the power consumption.

These new results are now included in the Results section. We also incorporated the methodology for obtaining them in the Methods section.

Figure 4. Energy to infer one image depending on the batch size and the number of sample. **a** For the Jetson’s GPU only. **b** For the overall system

Comment 3:

In this version, the authors removed the power estimation and instead provided typical power consumption from previous works, indirectly illustrating the power advantages of memristors and PCM chips. While this information is useful, it would be beneficial to include a more detailed analysis of the power efficiency in the revised manuscript.

Based on this comment, we included a more detailed power analysis in the revised paper. When looking at how power is distributed in fully integrated analog in-memory computing circuits, it is seen that a significant portion, is consumed by peripheral circuits, as documented, e.g., in ref.³. More specifically, operations at the neuron level, which include DAC, ADC, and activation functions, are the primary contributors to power consumption, accounting for approximately 40% of the total in ref.³. ADCs (Analog-to-Digital Converters) are in fact the most significant power consumers, accounting for even more than 40% in ref.⁴. The energy for charging the word lines of the array is also very significant, 30 to 40% of the total energy in ref.³. Consequently, although memristors have approximately five times lower resistance compared to PCM in the low-resistance state, as seen in Figure 3 of our main paper, this difference is not expected to have a significant impact on overall power consumption, as it has a minimal impact on both ADCs and word line charging energy.

These considerations are the reason for which we prefer to rely on these other works for our energy estimates. The fabricated in-memory computing circuit that we used in our work includes essential periphery circuitry on-chip (see its description in Supplementary Note 13), but the ADCs are off-chip. Its energy consumption is therefore not representative of the one of a final product.

We have edited the Discussion section of the paper to clarify these points.

Comment 4:

As mentioned in the rebuttal letter, I agree that the uncertainty evaluation task is more challenging than classification tasks. The slow modeling of uncertainty is a main drawback of BNNs, and if the authors' method can accelerate BNNs, it could be a potential solution. However, it is essential to compare the proposed solution with well-established deep learning algorithms and discuss their respective advantages and limitations. I suggest referencing a few works on this topic, such as "Bounding Box Regression with Uncertainty for Accurate Object Detection" (CVPR 2019) and "Gaussian YOLOv3: An Accurate and Fast Object Detector Using Localization Uncertainty for Autonomous Driving" (ICCV 2019), to provide a more comprehensive comparison.

Based on this recommendation, we added a comparison with the works suggested in the paragraph of the Discussion section positioning our work with regard to non-Bayesian approaches:

“Two principal methods exist for estimating uncertainty in non-Bayesian artificial neural networks (ANNs). The first, deep ensembles, trains multiple identical ANNs, creating a prediction distribution but offering no energy or hardware benefits⁵. Moreover, implementation challenges arise when transferring high-precision parameters into the imprecise conductance states of resistive memory in memristor-equipped ANNs. In contrast, Bayesian neural networks adeptly exploit memristor variability to store random variables, making them ideal for resistive memory-based hardware. The second method, Monte Carlo dropout, generates a prediction distribution by randomly disabling nodes within the model^{6,7}. However, the requirement for multiple forward passes precludes any reduction in energy consumption, and it is not natural to implement in a memristor-based circuit. Besides these general techniques, some task-specific approaches have also been proposed. In particular, some models use artificial neural networks representing Gaussian distributions, where one neuron represents the mean and another the standard deviation of a distribution. This approach has proven useful in tasks such as detecting out-of-distribution data in video surveillance or improving the accuracy of bounding box regression⁸⁻¹¹. Bayesian neural networks constitute a more general solution to the uncertainty evaluation challenge.”

References

1. Esmanhotto, E. *et al.* Experimental demonstration of multilevel resistive random access memory programming for up to two months stable neural networks inference accuracy. *Adv. Intell. Syst.* 2200145 (2022).
2. Harabi, K.-E. *et al.* A memristor-based bayesian machine. *Nat. Electron.* **6**, 52–63 (2023).
3. Wan, W. *et al.* A compute-in-memory chip based on resistive random-access memory. *Nature* **608**, 504–512 (2022).
4. Xue, C.-X. *et al.* A cmos-integrated compute-in-memory macro based on resistive random-access memory for ai edge devices. *Nat. Electron.* **4**, 81–90 (2021).
5. Lakshminarayanan, B., Pritzel, A. & Blundell, C. Simple and scalable predictive uncertainty estimation using deep ensembles. In *Advances in Neural Information Processing Systems 30 (NIPS 2017)* (2017).
6. Gal, Y. & Ghahramani, Z. Dropout as a bayesian approximation: Representing model uncertainty in deep learning. In *Proceedings of The 33rd International Conference on Machine Learning, PMLR 48:1050-1059, 2016* (2016).
7. Sida Wang, a. M. Fast dropout training. In *Proceedings of the 30th International Conference on Machine Learning, PMLR 28(2):118-126, 2013* (2013).
8. Kingma, D. P. & Welling, M. Auto-encoding variational bayes. *arXiv preprint arXiv:1312.6114* (2013).
9. Choi, J., Chun, D., Kim, H. & Lee, H.-J. Gaussian yolov3: An accurate and fast object detector using localization uncertainty for autonomous driving. In *Proceedings of the IEEE/CVF International conference on computer vision*, 502–511 (2019).
10. He, Y., Zhu, C., Wang, J., Savvides, M. & Zhang, X. Bounding box regression with uncertainty for accurate object detection. In *Proceedings of the IEEE/CVF conference on computer vision and pattern recognition*, 2888–2897 (2019).
11. Fan, Y. *et al.* Video anomaly detection and localization via gaussian mixture fully convolutional variational autoencoder. *Comput. Vis. Image Underst.* **195**, 102920 (2020).

Review on the revised manuscript “**Bringing uncertainty quantification to the extreme-edge with memristor-based Bayesian neural networks**” authored by Djohan Bonnet *et al.*, submitted to *Nature Communications*. [Manuscript Number: NCOMMS-23-00214B]

This reviewer appreciates the responses to the comments, and the authors incorporated them into the manuscript and figures appropriately. The revised paper is qualitatively improved over the initial version. Therefore, this paper is suitable for publication in this journal after addressing the typos and grammar errors.

- In supplementary note 3, ‘Distinct clusters of data points represents different programming conditions.’ → ‘Distinct clusters of data points represent different programming conditions.’ and ‘In the case phase change memories ~~’ → ‘In the case of phase change memories ~~’
- The first paragraph of the text below the supplementary Fig. 6, ‘However, as Supplementary Note 7 dmonstartes ~~’ → ‘However, as Supplementary Note 7 demonstrates ~~’
- The second paragraph of the text below the supplementary Fig. 6, ‘This results confirm that the mean value of conductance distributions remain ~~’ → ‘This results confirm that the mean value of conductance distributions remains ~~’
- In supplementary note 6, ‘In this simulator, the synaptic weight are ~~’ → ‘In this simulator, the synaptic weights are ~~’
- In the caption of supplementary Fig. 8b, ‘The performances are evaluated for ten training run.’ → ‘The performances are evaluated for ten training runs.’

Reviewer #3 (Remarks to the Author):

I appreciate the authors' extensive efforts on additional experiment and revision of the manuscript. The relays well addressed most of my main concerns. However, I noticed minor typographical errors within the supplementary materials, such as the instance in Note 14 (“an an analog”). I highly recommend a meticulous proofreading to keep the manuscript's quality. Overall, I have no more technical questions and recommend the manuscript for publication in Nature Communications.

Response to Reviews

Once again we would like to thank the reviewers whose comments have continually led us to improve the strength of the manuscript. We are delighted with their positive evaluation of our most recent work as well as our current version of the paper. We have addressed the typo and grammar errors raised by the reviewers in the Supplementary material.

Reviewer 1 (Remarks to the Author)

General comments

This reviewer appreciates the responses to the comments, and the authors incorporated them into the manuscript and figures appropriately. The revised paper is qualitatively improved over the initial version. Therefore, this paper is suitable for publication in this journal after addressing the typos and grammar errors.

- In supplementary note 3, ‘Distinct clusters of data points represents different programming conditions.’ → ‘Distinct clusters of data points represent different programming conditions.’ and ‘In the case phase change memories ’ → ‘In the case of phase change memories ’
- The first paragraph of the text below the supplementary Fig. 6, ‘However, as Supplementary Note 7 dmonstartes ’ → ‘However, as Supplementary Note 7 demonstrates ’
- The second paragraph of the text below the supplementary Fig. 6, ‘This results confirm that the mean value of conductance distributions remain ’ → ‘This results confirm that the mean value of conductance distributions remains ’
- In supplementary note 6, ‘In this simulator, the synaptic weight are ’ → ‘In this simulator, the synaptic weights are ’
- In the caption of supplementary Fig. 8b, ‘The performances are evaluated for ten training run.’ → ‘The performances are evaluated for ten training runs.’

We have corrected the typos and grammar errors in the Supplementary material.

Reviewer 3 (Remarks to the Author)

General comments

I appreciate the authors’ extensive efforts on additional experiment and revision of the manuscript. The relays well addressed most of my main concerns. However, I noticed minor typographical errors within the supplementary materials, such as the instance in Note 14 (“an an analog”). I highly recommend a meticulous proofreading to keep the manuscript’s quality. Overall, I have no more technical questions and recommend the manuscript for publication in Nature Communications. We have

proofread the Supplementary material and corrected several typos and grammar errors.